# Log-normal Mutations and their Use in Detecting Surreptitious Fake Images

**Ismail Labiad**                                    *ilabiad@meta.com*
*Meta, FAIR*

**Thomas Bäck**                          *t.h.w.baeck@liacs.leidenuniv.nl*
*LIACS, Leiden University, The Netherlands*

**Pierre Fernandez**                                    *pfz@meta.com*
*Meta, FAIR*

**Laurent Najman**                          *laurent.najman@esiee.fr*
*Univ Gustave Eiffel, CNRS, LIGM, France*

**Tom Sander**                                    *tomsander@meta.com*
*Meta, FAIR*

**Furong Ye**                                *f.ye@liacs.leidenuniv.nl*
*LIACS, Leiden University, The Netherlands*

**Mariia Zameshina**                      *mariia.zameshina@esiee.fr*
*Univ Gustave Eiffel, CNRS, LIGM, France*

**Olivier Teytaud**                                *oteytaud@meta.com*
*Meta, FAIR*

**Reviewed on OpenReview:** *https://openreview.net/forum?id=0RJvZY0h60*

## Abstract

In many cases, adversarial attacks against fake detectors employ algorithms specifically crafted for automatic image classifiers. These algorithms perform well, thanks to an excellent ad hoc distribution of initial attacks. However, these attacks are easily detected due to their specific initial distribution. Consequently, we explore alternative black-box attacks inspired by generic black-box optimization tools, particularly focusing on the log-normal algorithm that we successfully extend to attack fake detectors. Moreover, we demonstrate that this attack evades detection by neural networks trained to flag classical adversarial examples. Therefore, we train more general models capable of identifying a broader spectrum of attacks, including classical black-box attacks designed for images, black-box attacks driven by classical optimization, and no-box attacks. By integrating these attack detection capabilities with fake detectors, we develop more robust and effective fake detection systems.

## 1 Introduction and outline

Due to the recent progress in AI-based image generation(Rombach et al., 2022), now easily available on internet, fake images (i.e. images created by artificial intelligence) are invading social networks. Fortunately, various fake image detectors (fake detectors), based on watermarking or supervised fake detection, have been proposed for detecting such fake images.

While new fake detectors are proposed, we also have more and more attacks targeted to make fake detectors fail. In the context of fake image detection, adversarial attacks such as Square Attack (SA) (Andriushchenko

et al., 2020) or no-box techniques like diffusion-based purifiers (Saberi et al., 2024a) can be used to subtly alter fake images while enabling them to effectively bypass certain fake detectors. Black-box attacks(Guo et al., 2019; Biggio et al., 2013) are imperceptible modifications of an image aimed at deceiving a classifier: typically, a black-box attacked fake image becomes undetected by a fake detector based on supervised machine learning. No-box attacks are different: they have no prior knowledge on the model, and proceed by denoising: they typically remove watermarks, and therefore they deceive detectors based on watermarks. Therefore, detecting these attacks is necessary for enhancing the reliability of fake detectors. In the present paper, we analyze lesser known attacks inspired by the black-box optimization community: their algorithms perform well, and are sufficiently different for being undetected by detectors trained on classical attacks only[1].

In order to increase the diversity of black-box attacks and make detectors more robust, we consider attacks based on generic black-box optimization methods, in particular evolutionary methods. Over the past decades, iterative optimization heuristics like Evolutionary Algorithms (EAs) have proven effective in tackling challenging optimization problems. While specialized methods have emerged for specific domains—such as Genetic Algorithms for discrete optimization and Evolution Strategies (ES) for continuous optimization—various adaptations of these techniques have been proposed and successfully applied across diverse domains. Regarding the topic of *borrowing ideas across different domains*, there exists much work in exchanging ideas between discrete and continuous optimization domains. For example, variants of Covariance Matrix Adaptation ES (CMA-ES) and Differential Evolution (DE) have been proposed for specific problems (Hansen & Ostermeier, 2003; Hamano et al., 2022; Das et al., 2016). More recently, a study has investigated the ways of discretizing CMA-ES and its performance on different discrete BBOB problems (Thomaser et al., 2023). In this work, we investigate the advantage of borrowing an idea from a self-adaptive pseudo-Boolean optimization algorithm to the continuous domain. One of EAs' key algorithmic components is the utilization of a probability distribution for generating new search points, including an iterative adaptation of this distribution based on the objective function values of the newly generated solutions (Doerr et al., 2020; Doerr & Neumann, 2020). A wide range of such update strategies has been proposed over the past decades, for example, for multivariate Gaussian distributions in the context of covariance matrix adaptation in evolution strategies for continuous optimization problems (Hansen & Ostermeier, 2003) and for binomial and a variety of other mutation strength distributions in the context of mutation rates in evolutionary algorithms for pseudo-Boolean optimization (Doerr et al., 2020). Among those adaptation methods for the mutation strength in pseudo-Boolean optimization, the so-called **log-normal mutation** was developed almost thirty years ago as a method for allowing the mutation strength to quickly shift from exploration to exploitation (Bäck & Schütz, 1995; Bäck & Schütz, 1996; Kruisselbrink et al., 2011). As shown in (Doerr et al., 2020), on a set of 23 pseudo-Boolean test functions, log-normal mutation shows an empirical cumulative distribution functions (ECDF) performance across all 23 functions that are very close to the best-performing algorithm. In this paper, we compare the log-normal (LN) mutation with various algorithms for continuous optimization problems selected from the Nevergrad benchmarks (Rapin & Teytaud, 2018). Experimental results indicate that the log-normal strategy is competitive for problems that are particularly difficult, i.e., multi-modal and highly-deceptive. Inspired by the mentioned benchmarking results, we test the log-normal mutation for a practical scenario, namely the attack of fake detectors.These methods are not necessarily better than e.g. SA (Andriushchenko et al., 2020): but they are completely different, not specialized on images, and free from typical artifacts from specialized methods (Fig. 5.1), making them undetected by existing detectors. We then propose new detectors, for mitigating the risk associated to those methods based on generic black-box optimization.

**Outline:** Section 2 presents the state of the art in fake detectors (section 2.1) and Black-Box Optimization (BBO, section 2.2). Section 3 present our tools: section 3.1 for the log-normal mutations and section 3.2 for the extension of log-normal to continuous domains. Then section 4 presents preliminary experimental results, including an ablation (section 4.1), results on the Nevergrad benchmark suite (Section 4.2) and discussing specific results (Section 4.3): these preliminaries justify the choice of LogNormal as a tool for our black-box attacks of fake detectors, satisfying both (i) reasonably effective (ii) completely different from usual attacks. Section 5 focuses on the application to fake detectors: comparing various algorithms on this application, we

---

[1]All data collection and experiments were conducted on university servers by people with corresponding affiliations.

observe particularly good performance of log-normal mutation. Section 6 combines our new detectors and existing ones for improved performance.

**Contributions:** (1) We present an extension of log-normal mutation (usually applied in the context of discrete domains) to arbitrary discrete, continuous and mixed search spaces (Section 3.2). (2) We present a comparison between log-normal mutation and a state-of-the-art algorithm selector (Meunier et al., 2022), illustrating a particular strength of log-normal for deceptive and multimodal problems (Section 4.2, 4.3). (3) We show that log-normal mutation is particularly well suited for attacking fake image detectors: log-normal mutation performs well (Section 5.1), and the created fake images go undetected by detectors based on existing attacks (Section 5.2.2). For converting the attacks into defense, we then add a detector of these attacks (Section 5.2.2), and study its robustness (Section 6).

## 2 State of the art

In this section, we briefly discuss the state-of-the-art in fake detection methods, to set the stage for using those as a specific application domain for black-box-optimization (section 2.1). Then, we introduce the general setup of the black-box optimization problem, and the set of algorithms that are used in the empirical comparison presented in this paper (section 2.2).

### 2.1 Fake detectors

Due to the many AI-generated images on internet, it becomes important to be able to detect them. DIRE (Wang et al., 2023) is a method to detect images generated by Latent Diffusion Models (LDM) (Rombach et al., 2022). It focuses on the error between an input image and its reconstruction counterpart using a pre-trained diffusion model. This method is based on the observation that the generated images can be approximately reconstructed by a diffusion model, while real images cannot. Several papers already mentioned artifacts making DIRE unreliable in a real-world scenario, in particular the impact of the image format on the prediction (Moskowitz et al., 2024). Another line of work (Coccomini et al., 2023) explores the task using simpler, more traditional machine learning algorithms like Multi-Layer Perceptrons (MLPs), features extracted by Contrastive Language-Image Pretraining (CLIP, (Radford et al., 2021)), or traditional Convolutional Neural Networks (CNNs). Generalization to unseen image generators is known as a critical issue for such supervised learning approaches: Universal Fake Detector (Ojha et al., 2023) uses a linear model trained on top of a generic feature extractor, for the sake of generalization and transfer, and GenDet (Zhu et al., 2023) uses an adversarial method aimed at solving the problem of unknown fake generators. Another method that helps to detect LDM-generated images is the Local Intrinsic Dimensionality (LID) method (Lorenz et al., 2023). LID is employed to estimate the intrinsic dimensionality of a learned representation space, measuring the average distance between a point and its neighboring points. This approach is vital in characterizing the distinct properties of adversarial and natural samples in the latent space of a classifier. The multiLID method (Lorenz et al., 2023) extends this concept, combining locally discriminative information about growth rates in close proximity, which proves effective in detecting adversarial examples as well as images generated by diffusion models. Watermarking (Cox et al., 2007) and on-device certification (Nagm et al., 2021) are alternative methods for distinguishing between fake and real images. Watermarking is designed to create imperceptible alterations to the images. These alterations, however, can be detected and decoded with the appropriate algorithms. The primary purpose of watermarking is to ensure traceability of the images back to the model. Many of the current generation models already use built-in watermarking techniques. For instance, in Stable Diffusion (Rombach et al., 2022) the authors use Discrete Wavelet Transform (DWT) together with Discrete Cosine Transform (DCT) (Al-Haj, 2007). While these transformations provide some protection from minor generative image alterations, they are not resistant, for example, to image resizing. Furthermore, one can easily disable the watermarking in the Stable Diffusion code. More advanced watermarking methods, as proposed, for example, by Fernandez et al. (2023), merge watermarking into the generation process. Watermarking methods are brittle to adversarial attacks. They include no-box attacks which do not need any knowledge about the method, e.g., using perceptual auto-encoders (Fernandez et al., 2023) or noising/denoising with a diffusion model (Nie et al., 2022); and white-box or black-box attacks (in particular, SA) which employs the watermark extractor or an API access to it (Jiang et al.,

2023). Fortunately, there are effective detectors for both no-box attacks and for SA. We investigate how generic black-box optimization methods can be turned into adversarial attacks that are not detected by these detectors: therefore, we need detectors for this family of attacks.

## 2.2 Black-box optimization

An unconstrained optimization problem can be generally formulated as follows, $\min f(x), \ x \in \Omega$. where $\Omega$ denotes the search space, $f : \Omega \to \mathbb{R}^m$, and $m$ is the number of objectives. Note that we consider $m = 1$ in this work, and without loss of generality, we assume a minimization problem. According to the domain of $\Omega$, we can consider problems as continuous optimization when $\Omega \subset \mathbb{R}^n$, as discrete optimization when $\Omega \subset \mathbb{Z}^n$, and pseudo-Boolean optimization (PBO) when $\Omega = \{0, 1\}^n$, where $n$ is the dimensionality of the problem. We deal with black-box optimization (BBO), in which algorithms can not obtain the exact definition of the objective function $f$ and constraint definitions regarding the structure of $f$. Evolutionary computation has been widely applied to solve BBO. For example, Evolution Strategies (ES), Differential Evolution (DE), etc., have achieved success in solving continuous BBO problems (Bäck et al., 2023). The variants of these methods have also been applied for discrete BBO based on relaxation to a continuous problem (Pan et al., 2008; Hamano et al., 2022). Moreover, EAs have been well-studied for pseudo-Boolean optimization (PBO) (Doerr et al., 2020). Some strategies have been commonly applied with specific adjustments when solving different types of optimization problems. A recent study has investigated the performance of a discretized Covariance Matrix Adaptation Evolution Strategy (CMA-ES), addressing the adaptation of continuous optimization algorithms for the discrete domain (Thomaser et al., 2023).

In this work, we work on continuous optimization by utilizing discrete optimization algorithms. Specifically, we investigate utilizing the techniques of the $(1+\lambda)$ EAs (where $\lambda$ refers to the parallelism) that are designed for PBO for continuous optimization (Doerr et al., 2020). The $(1 + \lambda)$ EAs flip a number $\ell$ of variables to generate $\lambda$ new solutions from a single parent solution iteratively, and self-adaptive methods have been proposed to adjust the value of $\ell$ online, essentially detecting the optimal number of variables to be altered dynamically. We consider various black-box optimization methods, which can all be found in (Rapin & Teytaud, 2018). We include many algorithms, specified in Section D.1. We selected, for readability, a sample of methods covering important baselines (such as random search), one representative method per group of related methods, and the overall best methods. We refer to (Rapin & Teytaud, 2018) for the details about other algorithms, and to Beyer (2001) and references in Section D.1 for more details about generic optimization algorithms.

## 3 Introducing the LogNormal method as a different black-box method

The present section introduces the LogNormal algorithm and its adaptation to continuous settings. The next section will then validate it experimentally, so that Sections 5 and 6 will use the LogNormal method (and corresponding defenses) in our context of fake detection.

### 3.1 Log-normal mutations

We introduce in this section the general framework of the $(1 + \lambda)$ EA with log-normal mutation, which was proposed for the pseudo-Boolean optimization task, and illustrate a straightforward generalization of this algorithm to continuous and integer domains.

The log-normal mutation was first described in (Bäck & Schütz, 1995; Bäck & Schütz, 1996), later refined in (Kruisselbrink et al., 2011). It stands out for its robust performance across various problem landscapes (Doerr et al., 2020). Although it may not surpass other self-adjusting methods, such as the two-rate (Doerr et al., 2019b) and the normalized bit mutation (Ye et al., 2019), which are good at converging to small mutation strength $\ell = 1$ for the classic theory-oriented problems OneMax, LeadingOnes, etc., log-normal mutation consistently delivers competitive results for complex practical problems such as Ising models and Maximum Independent Vertex Set (Doerr et al., 2020). Following the setup in recent work (Doerr et al., 2020) on pseudo-Boolean optimization, we provide a concise formulation of a $(1 + \lambda)$ EA with the log-normal mutation in Algorithm 1. The algorithm starts from a randomly initialized point (line 2) and generates $\lambda$ offspring by the

---

**Algorithm 1:** $(1 + \lambda)$ $\text{EA}_{log-n}$ with log-normal mutation.

---

**1** **Input:** A given problem $f : \Omega \to \mathbb{R}$, where $n$ denotes the dimensionality of the problem, an initial value of the mutation rate $p \in (0, 1)$, population size $\lambda > 0$, and a learning rate $\gamma = 0.22$;

**2** **Initialization:** Sample $x \in \Omega$ u.a.r. and evaluate $f(x)$;

**3** **Optimization: for** $t = 1, 2, 3, \dots$ **do**

**4**     **for** $i = 1, \dots, \lambda$ **do**

**5**        $q \sim \mathcal{N}(0, 1)$;

**6**        $p^{(i)} = \left(1 + \frac{1-p}{p} \cdot \exp(\gamma \cdot q)\right)^{-1}$ ;

**7**        Sample $\ell^{(i)} \sim \text{Bin}_{>0}(n, p^{(i)})$;

**8**        create $y^{(i)} \leftarrow \text{MUTATE}(\ell^{(i)}, x)$; evaluate $f(y^{(i)})$ ;               `// See Alg.2`

**9**     $i \leftarrow \min \left\{ j \mid f(y^{(j)}) = \max\{f(y^{(k)}) \mid k \in [\lambda]\} \right\}$;

**10**     $p \leftarrow p^{(i)}$;

**11**     $x^* \leftarrow \arg\max\{f(y^{(1)}), \dots, f(y^{(\lambda)})\}$ ;        `// ties broken by favoring the smallest index`

**12**     **if** $f(x^*) \leq f(x)$ **then** $x \leftarrow x^*$;

**13** **Output:** $x, f(x)$

---

**Algorithm 2:** $\text{MUTATE}(\ell, x)$. Note that the uniform random distribution is applied to the domain of the corresponding variable: in the case of our attacks (which are always bounded for $L^\infty$), this means randomly drawing in $[-0.03, 0.03]$. Similarly, the general purpose benchmarks in appendix, the mutation uses the domain of the corresponding variable in the corresponding problem.

---

**1** **Input:** a solution $x \in \bigtimes_{i=1}^{n} \mathcal{X}_i$, and the mutation strength $\ell \in \{1, 2, \dots, n\}$, where $n$ is the dimensionality of the problem;

**2** Sample $\ell$ pairwise different positions $i_1, \dots, i_\ell \in [n]$ u.a.r.;

**3** $y \leftarrow x$;

**4** **for** $j = 1, \dots, \ell$ **do**

**5**     **repeat**

**6**        $k \sim \mathcal{U}(\mathcal{X}_{i_j})$ ;                   `// Sample k u.a.r. from domain` $\mathcal{X}_{i_j}$

**7**     **until** $k \neq x_{i_j}$;

**8**     $y_{i_j} \leftarrow k$;

**9** **Output:** $y$;

---

MUTATE function in the for-loop (lines 4-8). We denote the mutation strength $\ell$ as the number of variables to be altered, and this number is generated for each newly generated solution candidate, i.e., offspring, by first mutating the mutation rate $p$. In practice, $p$ is mutated according to a log-normal distribution (line 6, where $\mathcal{N}(0, 1)$ is a normally distributed random variable with expectation zero and standard deviation one, from which $q$ is sampled in line 5), and this rule maintains that the median of the distribution of new mutation rates $p^{(i)}$ is equal to the current mutation rate $p$. $p^{(i)}$ then determines the value of $l^{(i)}$ (line 7), and the corresponding offspring is created and evaluated in line 8. Specifically, $\ell^{(i)}$ is sampled from a binomial distribution $\text{Bin}_{>0}(n, p^{(i)})$ (line 7), where $n$ is the dimensionality, and the sampling is repeated until obtaining $\ell^{(i)} > 0$. In case of ties in the objective function values, the mutation rate used for the first best of the newly generated solutions (line 9) is taken for the next iteration (line 10). The best solution $x^*$ is updated in line 12 if the best offspring is equal to or better than the current parent solution.

### 3.2 Modifications of the original log-normal algorithm: adaption to the continuous setting

Note that in Algorithm 1 we do not specify the domain $\Omega$ of $x$. The log-normal mutation has been commonly applied for the pseudo-Boolean optimization, i.e., $\Omega = \{0, 1\}^n$, and the MUTATE-operator flips $l^{(i)}$ bits that are selected uniformly at random. In this work, we introduce a generalization of the MUTATE-operator, as presented in Algorithm 2, to extend the log-normal mutation to other domains. This generalized MUTATE is

| Name | Initial $p$ | $\lambda$ | Name | Initial $p$ | $\lambda$ |
|---|---|---|---|---|---|
| Lognormal (Standard setting) | 0.2 | 12 | SmallLognormal | 0.2 | 4 |
| BigLognormal | 0.2 | 120 | XLognormal | 0.8 | 12 |
| HugeLognormal | 0.2 | 1200 | XSmallLognormal | 0.8 | 4 |
| OLN(Combined with bandits) | 0.2 | 12 | | | |

Table 1: Variants of log-normal mutations used in the experiments. OLN is a combination with bandits, for managing reevaluations in a noisy optimization context.

applicable for arbitrary search domains $\Omega = \bigtimes_{i=1}^{n} \mathcal{X}_i$, where $\mathcal{X}_i \in \{\{0,1\}, \mathbb{Z}, \mathbb{R}, \mathbb{N}\}$. It samples a new value that is distinct from the current one at random from the respective domain for each of $\ell$ variables, which are selected (uniformly at random) for mutation. This technique has also been adopted by Nevergrad (Rapin & Teytaud, 2018), which automatically adapts algorithms designed for continuous domains to also work for discrete domains and vice versa.

### 3.3 The LogNormal attack applied to fake detectors: difference with Square Attacks

In the present Section, we clarify what the LogNormal attack is doing on an image. The adversarial attack consists in finding $\epsilon$ such that $I + \epsilon$ is classified as non-fake, whereas $I$ is classified as fake. We maintain $-0.03 \leq \epsilon_{i,j,k} \leq 0.03$ (corresponding to pixel $(i,j)$ for channel $k$), i.e., our attack is bounded in $L^\infty$ the same way as SA and other attacks. Note that the naming LogNormal corresponds to the tool used by the algorithm for choosing the mutation rate (as detailed in Section 3.1): this does not imply that the noise added to the image follows a log-normal probability distribution. As explained in Section 3.2, the perturbation affects a subset of pixels, with an added value $\epsilon_{i,j,k}$ randomly uniformly drawn in $[-0.03, 0.03]$. Compared to SA, note that the algorithm does not generate the typical horizontal or vertical artifacts corresponding to the squares (Fig. 5.1): this explains that it is more difficult to identify, both by supervised learning and by human investigation.

## 4 Preliminaries: experimental results on the Nevergrad Black-box optimization benchmarks

The present section shows that the LogNormal algorithm performs reasonably well, including in the unexpected continuous settings, in particular for the black-box optimization benchmarks from Nevergrad that are somewhat (as far as possible for generic benchmarks unrelated to vision) related to our image context.

### 4.1 Parameter settings & Ablation

While the log-normal mutation can control mutation rates of the $(1+\lambda)$ EA online, Algorithm 1 still comprises three hyperparameters, i.e., the initial value of $p$, population size $\lambda$, and $\gamma$, that can affect the algorithm's performance. We set $\gamma = 0.22$ following the suggestion in previous studies (Kruisselbrink et al., 2011; Doerr et al., 2020). For the other two hyperparameters, we test two values 0.2 and 0.8 for $p$ and various population sizes $\lambda$. The detailed combinations are presented in Table 1.

We denote log-normal as the standard setting. BigLognormal experiments are performed with an increase in population size (respectively, large increase for HugeLognormal, and decrease for SmallLognormal). XLognormal obtains a greater initial mutation rate. OLN (Optimistic Lognormal) experiments the combination with Optimism as performed in Nevergrad for making deterministic algorithms compatible with noisy optimization (see appendix D.3).

We compare the settings of Algorithm 1 with several algorithms, such as random search, CMA, and the anisotropic adaptive algorithm (Doerr et al., 2016; 2017) provided by Nevergrad. log-normal with the standard parameterization from (Bäck & Schütz, 1996) performs essentially well across the 14 tested problems (see Appendix F), though the "Big" variant is an interesting outsider. We note that the default parametriza-

tion of log-normal fails mainly on topology optimization (for which anisotropic methods perform great) and on noisy problems (which are irrelevant for our context of attacking deterministic fake detectors, and we note that a good solution in that setting is to use a combination with optimism in front of uncertainty as proposed in (Rapin & Teytaud, 2018)). We therefore keep the standard log-normal and some "Big" counterparts for our application to fake detectors.

## 4.2 Robust performance of log-normal

The log-normal algorithm has already shown competitive results in existing benchmark studies (Doerr et al., 2019b), and in the present paper we examine its performance by comparing it on extensive benchmarks provided by Nevergrad. Table 11 shows the diversity of the benchmarks in Nevergrad, where each benchmark is accompanied with a list of baselines.

Since NGOpt is a "wizard" provided by Nevergrad that is tuned for selecting automatically a proper algorithm for each problem, we use it as a baseline for comparison. We run log-normal and NGOpt and proposed default methods.Tables 2 and 12 list the rank of the log-normal algorithm and of NGOpt: Table 2 lists the problems in which the benchmark outperforms NGOpt, and Table 12 presents the results of the other benchmarks.

We observe that the log-normal algorithm can outperform NGOpt on 17 out of 41 benchmarks and perform better than 50% of all algorithms on 29 benchmarks. Recall that Nevergrad contains a diverse set of benchmarks, and it is well-known that we can not expect one algorithm to perform the best for all problems. The lognormal algorithm shows robust and competitive performance for the single-objective, difficult, low ratio budget/dimension benchmarks in Nevergrad. Of course, NGOpt is difficult to beat: it is an optimization wizardLiu et al. (2020), automatically selecting a black-box optimization method specifically for the dimension, budget and type of variables: LogNormal can not compete in the general case as it is very naive for multi-objective, constrained or stochastic contexts: this is not our purpose, we just observe that LogNormal performs well on problems somewhat related to adversarial attacks.

Since the log-normal algorithm was originally proposed for discrete optimization, we first present the detailed results of log-normal variants for the two discrete benchmarks in Fig. 4. In Fig. 4, the y-axis represents the loss of algorithms for tested budgets (presented on the x-axis), and algorithm labels are annotated by their average loss for all the maximum budget between parentheses and (for checking stability) the average loss for the tested budgets excluding the largest one between brackets (if there are at least three budget values). For readability, we present only the 35 best results (and the worst, for scale) for each benchmark. We can observe that the log-normal variants obtain the best performance for the two discrete benchmarks.

## 4.3 Competitive results for deceptive problems

Recall that in the ranks in Table 2, the log-normal algorithm performs best for the "deceptive" and "fishing" benchmarks and outperforms NGOpt on many difficult multimodal benchmarks (marked with **). We plot in Fig. 5 the detailed results of the best of 61 algorithms on the deceptive benchmark. The deceptive benchmark combines many random translations of hard problems in many dimensionalities, including (i) a problem with a long path to the optimum, which becomes thinner and thinner close to the optimum, (ii) a problem with infinitely many local minima, (iii) a problem with condition number growing to infinity as we get closer to the optimum. We note that on this hard continuous benchmark, the log-normal algorithm performs well across all the tested budgets. Therefore, due to its robust performance across a number of benchmarks and particularly competitive performance for the hardest benchmarks, we utilize the log-normal mutation for the following fake detector scenarios.

## 5 New attacks of fake detectors

We attack fake detectors. More precisely, we add imperceptible noise $e$ to the image $x$, with this noise chosen so that fake detectors fail: typically $x$ is a fake image detected as fake by a detector $\mathcal{D}$ and we find a small $e$ such that $x + e$ is classified as non-fake by $\mathcal{D}$. Black-box algorithms can be straightforwardly applied to attacking a fake detector $\mathcal{D}$ (which returns the estimated probability $\mathcal{D}(x)$ that an image $x$ is fake) by

| Cases in which log-normal outperforms NGOpt | | | |
|---|---|---|---|
| Problem | Rank of log-normal | Num algorithms | Rank of NGOpt |
| deceptive** | 0 | 61 | 6 |
| fishing** | 0 | 63 | 56 |
| multiobj-example-many-hd | 1 | 67 | 48 |
| yatuningbbob | 4 | 93 | 54 |
| multiobj-example-hd | 5 | 69 | 55 |
| multiobj-example | 9 | 63 | 29 |
| yaonepensmallbbob | 9 | 79 | 15 |
| nano-seq-mltuning | 10 | 23 | 11 |
| yasmallbbob | 10 | 96 | 19 |
| nano-naive-seq-mltuning | 11 | 23 | 14 |
| zp-pbbob | 13 | 41 | 26 |
| nano-veryseq-mltuning | 14 | 26 | 16 |
| pbo-reduced-suite* | 16 | 156 | 20 |
| verysmall-photonics** | 26 | 64 | 30 |
| verysmall-photonics2** | 33 | 67 | 37 |
| pbbob | 35 | 63 | 42 |
| ultrasmall-photonics2** | 44 | 83 | 72 |

Table 2: Results on benchmarks from the Nevergard benchmarking suite for which log-normal outperforms NGOpt. The rank, computed as described in Section E, is between 0 and $num - algos - 1$. We compare log-normal to NGOpt. Cases in which log-normal is outperformed by NGOpt are in Table 12. We note that results on continuous problems are not bad, in particular for hard problems: Deceptive (which is designed for being hard), some multi-objective benchmarks (in particular the many-objective case), PBBOB which uses difficult distributions of random translations for optima, very low budget problems such as YaTuningBBOB, and difficult low budget Photonics or Fishing problems. * denotes discrete problems, ** denotes single-objective problems which are highly multimodal and difficult.

defining $loss(e) = \mathcal{D}(e + x)$ and minimizing $loss$ on a domain $D$ (typically $[-0.03, 0.03]^t$, where $t$ is the shape of the image tensor[2]) using a black-box algorithm. Our attacks (Section 5.1) consider the fake detector as a black-box, so we do not have gradients. Then, in Section 5.2, we create defense mechanisms by detecting various black-box attacks, showing that the log-normal attack is not detected by a detector created for a classical attack (such as Square Attack) and, therefore, needs new ad hoc detectors.

## 5.1 Black-box attacking fake detectors

The fake detector we consider is the universal fake detector (Ojha et al., 2023). Since the advent of adversarial attacks (Guo et al., 2019; Biggio et al., 2013), it is possible to modify data (e.g. an image) so that it is misclassified, while remaining, for visual inspection, close to the original image for the $L^\infty$ norm. This is critical for the success of fake detectors, as fake detectors can be attacked by such imperceptible adversarial attacks. Typical adversarial attacks for images include (Goodfellow et al., 2014; Biggio et al., 2013; Andriushchenko et al., 2020; Jiang et al., 2023; Chen et al., 2020; Jiang et al., 2023). Our dataset of fake images is sampled from the test set provided by the authors of Ojha et al. (2023). Precisely, we consider the datasets as described in Table 3. IN500- refers to the 500 first classes of ImageNet, IN500+ to the other classes of ImageNet, G refers to Glide, PG refers to ProGAN, BG refers to BigGAN, LDM-200 is defined in (Ojha et al., 2023). In the present paper, Clean means unattacked, neither by no-box nor by black-box attacks.

**Critical cases.** We consider the detection of attacks (no-box or black-box attacks) aimed at evading fake detectors. So, we are particularly interested in correctly detecting attacks in two cases (1) fake images attacked for making them appear genuine and (2) real, unaltered images (clean). The other cases are less

---

[2]In this case, we assume that pixels are in the range $[0, 1]$. Due to maximum values in image representations we might clamp the values of $e + x$.

| Name | Description |
|------|-------------|
| Dataset1 | 2k images from PG, 2k images from BG and 1k images from LDM-200 for a total of 5k fake images. Used to benchmark variations of log-normal attacks and square attack. |
| Dataset2-DP | Same as Dataset1, plus 5k real images from IN500- and their purified versions by DiffPure with parameter 0.1 |
| Dataset2-IR | Same as Dataset1, plus 5k real images from IN500- and their purified versions by ImageRephrase with parameter 0.1 |
| Dataset2-SA | Same as Dataset1, plus 5k real images from IN500- and their attacks by SA with budget=10k and $l^\infty = 0.01$. Total = 20k images. |
| Dataset2-LN | Same as Dataset1, plus 5k real images from IN500- and their attacks by log-normal (algo1) with budget=10k and $l^\infty = 0.01$. Total = 20k images. |
| Dataset3 | 1k real images from IN500+ + 500 fake images from PG of different classes than Dataset2 + 500 fake images from P and their attacked counterpart. Total = 4k images. |
| Dataset3-IR0.1, IR0.2, IR0.3 | Purified versions of Dataset3 based on ImageRephrase, with parameter 0.1, 0.2, 0.3. |
| Dataset3-DP0.1, DP0.2, DP0.3 | Purified versions of Dataset3 based on DiffPure, with parameter 0.1, 0.2, 0.3. |
| Dataset3-SA $(L,B)$ | Real images from Dataset3 + fake images from Dataset3 attacked by SA with various amplitudes ($L \in \{0.01, 0.03, 0.05\}$) and budget ($B \in \{1000, 10000\}$). |
| Dataset3-LN $(L,B)$ | Real images from Dataset3 + fake images from Dataset3 attacked by LN (algo1 to algo5) with various $L \in \{0.01, 0.03, 0.05\}$ and $B \in \{1000, 10000\}$. |
| Dtaset4 | 1k fake images from Dataset3, attacked with log-normal attacks (algo1). |

Table 3: Datasets used in the study. Dataset3 will be used as a test set for the detectors trained on Dataset2 and Dataset4, will be used for testing the transfer of detectors trained on SA attacks to log-normal attacks.

critical: (1) the case of real images attacked so that they will look fake: this will usually not prevent the original images from being classified as real, making the situation less annoying. (2) the case of clean fake images (erroneously viewing them as attacked will not prevent a fake detector from working). Therefore, besides our accuracies on datasets, we will present accuracies on the critical part of these datasets.

We first attack the fake detector with the classical SA, and then with generic black-box optimization methods from section 2.2. All our black-box attacks are bounded in $l^\infty$, and in all experiments, we use the same $l^\infty$ bound on attacks for all methods. We give a brief overview here and full details are in appendix D.2. Nevergrad provides modifiers dedicated to domains shaped as images. These modifiers can be applied to any black-box optimization algorithm. For example, the prefix *Smooth* means that the algorithm periodically tries to smooth its attack $x$, if the loss of $Smooth(x)$ is better than the loss of $x$, then $x$ is replaced by $Smooth(x)$ in the optimization algorithm. We add two additional modifiers specific from adversarial attacks. Both are inspired from SA, though our modifications have, by definition, less horizontal and vertical artifacts (Fig. 5.1). First, G (Great) means that we replace $loss(x)$ by $loss(0.03 \times \text{sign}(x))$ when the allowed norm of the attack is 0.03 for the $l^\infty$ norm. Second, SM (smooth) means that we replace the loss function $f$ by $f(convolve(x))$, where *convolve* applies a normal blurring with standard deviation 3/8. GSM means that we apply both G and SM. For example, we get GSM-SuperSmoothLognormalDiscreteOnePlusOne by applying G, SM and SuperSmooth as modifiers on top of the standard log-normal method, and various methods as

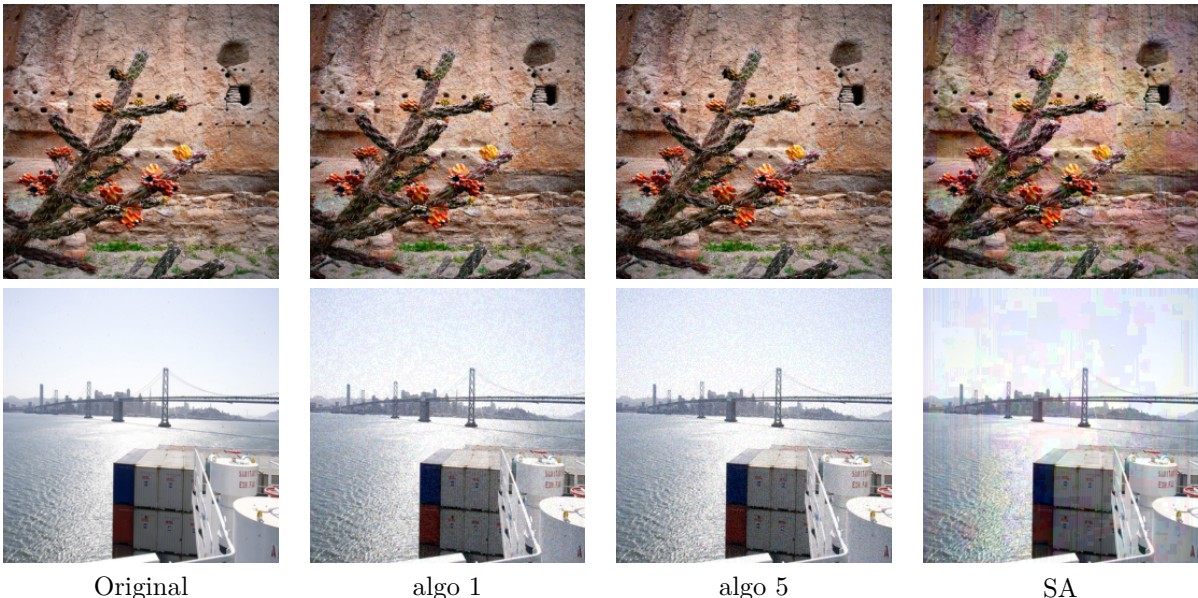

| Original | algo 1 | algo 5 | SA |

Figure 1: Example of attacked images. All attacks are done with a budget of 10k queries and $l^\infty = 0.03$. We observe vertical and horizontal artifacts in the case of SA.

| Algorithms | Alias | Algorithms | Alias |
|---|---|---|---|
| GSM-SuperSmoothLognormalDiscreteOnePlusOne | algo1 | LognormalDiscreteOnePlusOne | algo4 |
| G-SuperSmoothLognormalDiscreteOnePlusOne | algo2 | GSM-BigLognormalDiscreteOnePlusOne | algo5 |
| SuperSmoothLognormalDiscreteOnePlusOne | algo3 | G-BigLognormalDiscreteOnePlusOne | algo6 |

| Algorithm | SA | algo1 | algo2 | algo3 | algo4 | algo5 | algo6 |
|---|---|---|---|---|---|---|---|
| Success rate on dataset1 | 100% | 91.0% | 94.1% | 64.6% | 82.0% | 99.2% | 99.1% |

Table 4: Top: Variations of the log-normal algorithm considered. Bottom: Success rate when attacking the fake detector. The success rate is computed solely on attacks of correctly classified clean images. All attacks have 10k budget and $l^\infty = 0.03$. Inspired by SA, with maximum values of modifications ("G") covering an area ("SM"), Algo5 performs best among LogNormal variants. Compared to SA, it is less specific from images and it has a self-adaptive mutation rate starting at a high value ("Big"): the key point for us is that it performs reasonably well while the transfer from SA detectors fails.

in Tab 4 (top). Results are presented in Tab. 4 (bottom). Examples of attacked images are presented in Figure 5.1. Basically, when we have a black-box access to a fake detector, we can attack it either by SA or by log-normal with a budget of 10k queries and $l^\infty = 0.03$.

## 5.2 From attack to defense: detecting attacks

A recent trend is the denoising of watermarked images for evading the detection: whereas the watermark makes it possible to detect that an image is fake, the denoised version goes undetected. However, a defense is to detect such a denoising (e.g. by DiffPure or ImageRephrase), or other forms of attacks (such as SA). That way, our work both provides (i) new attacks and (ii) detectors for those attacks, to be used as an additional step (detection of evasion) in fake detectors.

| | | Attacks | | | | | |
|---|---|---|---|---|---|---|---|
| | Clean | DP 0.1 | DP 0.2 | DP 0.3 | IR 0.1 | IR 0.2 | IR 0.3 |
| FNR | 5.2% | 23.42% | 34.4% | 40.1% | 11.3% | 16.8% | 26.2% |
| PSNR | NA | 27.9 | 24.4 | 22.1 | 26.6 | 24.4 | 22.7 |

Table 5: The second row shows the false negative rate (FNR) at a threshold of 0.5 for the universal fake detector on clean Dataset1 consisting of only fake images and different purified versions of this dataset. DP stands for DiffPure while IR stands for ImageRephrase. The third row corresponds to the average PSNR.

### 5.2.1 No-box attacks (purifiers) and their detection

While watermarking techniques can be made robust against some image alterations such as resizing or JPEG compression as shown by Fernandez et al. (2023), they remain vulnerable to no-box purification attacks (Saberi et al., 2023) that destroy the hidden message. Additionally, the distribution change introduced by no-box purification can impact negatively the performance of a fake detector, as shown in Table 5. We will next show that one can easily detect these attacks if we have access to the purification method. We consider two purifiers used by Saberi et al. (2024a): DiffPure based on guided diffusion and ImageRephrase based on latent diffusion. We consider the strength/steps parameter in the set {0.1, 0.2, 0.3}. These two purifiers lead to an average PSNR (Peak Signal-to-Noise Ratio) between 20 and 30, which is a noticeable yet acceptable quality loss. As expected, the PSNR of the distortion increases with the strength parameter. For detecting the no-box attacks, we train a classifier, specified in table 14, on Dataset2-DP or Dataset2-IR. We experiement with both ResNet50 and SRnet (Boroumand et al., 2018) and the latter performs better overall. We use Dataset3 as a test set (so that the distributions of images differ) with images attacked by same purifier but with different parameters in $\{0.1, 0.2, 0.3\}$. The results of the detection of latent purifiers and guided diffusion purifiers are shown in Table 6 and Table 7. We achieve less than 5% FPR and FNR across the hold-out training dataset and the critical part of testset, although the FPR is relatively higher for the full testset that can be explained by having a sort of transfer from detecting DP or IR to detecting images generated by G. In short, training on a purifier with a specific parameter allows us to detect that same purifier with different parameters and for different image distributions.

### 5.2.2 The detection of black-box attacks

For detecting square attacks, we train a deep net using a similar setup, as the SRnet model still performs better than ResNet in this scenario. We also employ data augmentation techniques: horizontal flip, random crop and color jitter which empirically makes the model more robust to different attack parameters, while allowing us to train using only one attack parameter budget=10k and $l^{\infty} = 0.01$. The training is done on Dataset2-SA. For testing, we use Dataset3-SA using different parameters than those used during training. We then test the transfer of the SA detector on Dataset4, corresponding to Log-normal attacks. Table 8 summarizes the results of the SA detector, for detecting SA and as a detector of log-normal attacks. We observe that the transfer to detecting log-normal attacks is very poor. So, we need to include such attacks in our training for improving the defense.

**A new detector for log-normal attacks.** We have seen that detectors of SA do not detect our log-normal attacks. For detecting log-normal attacks, we use the same setup as in section 5.2.2 for creating a new detector. Dataset2-LN is used during training with the attacked images now obtained with GSM-SuperSmoothLognormalDiscreteOnePlusOne budget=10k and $l^{\infty} = 0.01$ and again, the dataset is split to 80% train, 10% test, 10% validation. For testing, we use Dataset3 with the images attacked using different variations of log-normal and parameters (Budget and $l^{\infty}$). Table 9 presents the results of the log-normal detector: we observe that the learning was made on images attacked by algo1 only and we get positive results for all log-normal variants.

| | Dataset | FPR↓ | FNR↓ | AUC↑ |
|---|---|---|---|---|
| (same dist) | Dataset2(param=IR0.1) hold-out | 0.9% | 0.3% | 0.99 |
| | Dataset 3, full (critical and non critical images) | | | |
| | Dataset3 (param=IR0.1) | 23.1% | 0.1% | 0.96 |
| (different dist) | Dataset3 (param=IR0.2) | 23.1% | 0.2% | 0.96 |
| | Dataset3 (param=IR0.3) | 23.1% | 0.2% | 0.95 |
| | Dataset 3, critical part | | | |
| | Dataset3 (param=IR0.1) | 1.7% | 0.0% | 0.99 |
| (different dist) | Dataset3 (param=IR0.2) | 1.7% | 0.0% | 0.99 |
| | Dataset3 (param=IR0.3) | 1.7% | 0.0% | 0.99 |

Table 6: False positive and false negative rates for purified images detection at a threshold of (0.5) alongside with the AUC score. The purified training images correspond to the latent-purifier (class "ImageRephrase") with parameter 0.1. We observe good results, in particular for critical cases.

| | Dataset | FPR↓ | FNR↓ | AUC↑ |
|---|---|---|---|---|
| (same dist) | Dataset2(param=DP0.1) hold-out | 1.9% | 3.9% | 0.99 |
| | Dataset 3, full (critical and non critical) | | | |
| | Dataset3 (param=DP0.1) | 23.0% | 3.7% | 0.88 |
| (different dist) | Dataset3 (param=DP0.2) | 23.0% | 4.0% | 0.87 |
| | Dataset3 (param=DP0.3) | 23.0% | 4.2% | 0.87 |
| | Dataset 3, critical part | | | |
| | Dataset3 (param=DP0.1) | 2.2% | 3.0% | 0.99 |
| (different dist) | Dataset3 (param=DP0.2) | 2.2% | 4.6% | 0.99 |
| | Dataset3 (param=DP0.3) | 2.2% | 5.4% | 0.99 |

Table 7: False positive and false negative rates for purified images detection at a threshold of (0.5) alongside with the AUC score. The purified training images correspond to the guided-diffusion purifier (class "Diff-Pure") with parameter 0.1. We observe good results, in particular when we restrict the analysis to critical cases (see the specification of critical cases in section 5.1).

## 6 Combining detectors

We need to detect both no-box and black-box attacks. As our experiments show that other black-box attacks could be used instead of classical ones, we also need transfer among different black-box attacks. Table 10 presents a combination of detectors: we classify an image as fake if it has an estimated probability of fake/attacked greater than $\frac{1}{2}$ for at least one of our base detectors. Our base detectors are (i) the detector of SA (ii) the detector of CMA attacks (iii) the detector IR (ImageRephrase (Saberi et al., 2024b)) (iv) the detector MBT (Minnen et al., 2018) (v) the watermark detector (vi) the UFD. On purpose, we do not include the detector of LogNormal attacks for checking that our combined detectors transfers to detecting LogNormal attacks without having seen them.

## 7 Conclusions

We tested log-normal mutations on various benchmarks and extended it to continuous benchmarks, including fake detection tasks. Lognormal mutations perform well in some cases in the Nevergrad benchmarks, especially on: (1) PBO, a classical discrete benchmark, and some other discrete benchmarks (2) In some continuous contexts, in particular in the most difficult scenarios such as many-objective, highly multimodal, low budget benchmarks, including e.g. real world benchmarks in photonics. We note that other discrete algorithms adapted to the continuous case do perform well. We note that discrete optimization methods are relevant in continuous optimization, when the prior (i.e. the range of reasonable values for each variable) is

| | Dataset | FPR↓ | FNR↓ | AUC↑ |
|---|---|---|---|---|
| (same dist) | Dataset2-SA hold-out | 1.8% | 0.5% | 0.99 |
| (different dist) | Dataset3-SA (B=10k, L=0.01) | 0.8% | 0.9% | 0.99 |
| | Dataset3-SA (B=10k, L=0.03) | 0.8% | 0.0% | 0.99 |
| | Dataset3-SA (B=10k, L=0.05) | 0.8% | 0.0% | 0.99 |
| | Dataset3-SA (B=1k, L=0.01) | 0.8% | 0.0% | 0.99 |
| | Dataset3-SA (B=1k, L=0.03) | 0.8% | 0.0% | 0.99 |
| | Dataset3-SA (B=1k, L=0.05) | 0.8% | 0.0% | 0.99 |
| (transfer to log-normal) | Dataset4 | NA | 79.5% | NA |

Table 8: False positive and false negative rates for square attack detection at a threshold of (0.5) alongside with the AUC score. B stands for the budget used for the attack and L stands for the accepted $l^\infty$ distance. Last row: (clearly failed) transfer to log-normal detection.

| | Dataset | FPR↓ | FNR↓ | AUC↑ |
|---|---|---|---|---|
| (same dist) | Dataset2-LN hold-out | 0.7% | 2.1% | 0.99 |
| (different dist) | Dataset3-LN (algo1, B=10k, L=0.01) | 4.1% | 5.4% | 0.98 |
| | Dataset3-LN (algo1, B=10k, L=0.03) | 4.1% | 2.3% | 0.99 |
| | Dataset3-LN (algo1, B=10k, L=0.05) | 4.1% | 2.9% | 0.99 |
| | Dataset3-LN (algo2, B=10k, L=0.03) | 4.1% | 4.1% | 0.98 |
| | Dataset3-LN (algo3, B=10k, L=0.03) | 4.1% | 11.7% | 0.95 |
| | Dataset3-LN (algo4, B=10k, L=0.03) | 4.1% | 1.6% | 0.99 |
| | Dataset3-LN (algo5, B=10k, L=0.03) | 4.1% | 0.4% | 0.99 |

Table 9: False positive and false negative rates for log-normal detection at a threshold of (0.5) alongside with the AUC score. Algorithms are reported in Table 4, B stands for the budget used for the attack and L stands for the accepted $l^\infty$ distance. LN stands for log-normal.

important and excellent precision is impossible (Sections 4.2 and 4.3). Of course this can not compete with classical local optimization in terms of convergence rates for large budgets, but we show that it can be great for a low ratio budget/dimension.

Fake detection presents a complex challenge, with the difficulty of identification varying based on factors such as the type of data and the model used (Epstein et al., 2023; Sha et al., 2023). We consider the setting in (Wang et al., 2023). We observe that log-normal mutations in continuous settings, as well as other generic black-box optimization algorithms, are credible attack mechanisms, so that detecting attacks is necessary for a good defense. We observe a very poor transfer between the detectors of different types of attacks and in particular from the detection of classical attacks (such as SquareAttack) to our modified attacks (such as log-normal), so that a good defense mechanism (aimed at improving the detection of manipulated images) must include the learning of diverse attack mechanisms.

We also note that detecting no-box attacks is a simple yet effective protection against them: we get a detection rate close to 100%, so that no-box attacks are vastly mitigated. Combining (simply by if-then-else) many tools can therefore lead to better fake detectors: classical fake detectors, watermarking detectors, no-box detection as proposed in the present paper, detectors of classical attacks such as SA, and detectors for our proposed black-box attacks based on generic black-box optimization tools.

Our most immediate further work is the combination, by machine learning instead of if-then-else, of our fake detectors (no-box, SA, lognormal, and existing detectors such as (Ojha et al., 2023) and (Fernandez et al., 2023), into a single detector. We also plan to. add new attacks and defenses based on generic black-box optimization, and a third work is the investigation of log-normal mutations (and other discrete algorithms) for other continuous problems.

Table 10: Performance of detectors on clean images, images under no-box attacks, and images under black-box attacks. Stable Signature p-value threshold is set to $10^{-2}$. The first column corresponds to the TNR (true negative rate), the other columns correspond to various forms of TPR (true positive rates).

| Detectors | Clean | | | No-box on watermarked | | Black-box on Fake | |
|---|---|---|---|---|---|---|---|
| | Real | Fake | Water-marked | IR | MBT | GSM Big-LN | SA |
| UnivFD | **98.1** | **99.9** | 20.8 | 53.9 | 24.4 | 3.4 | 0.0 |
| StableSignature | **99.9** | 0.0 | **100.0** | 39.2 | 45.6 | 0.5 | 0.4 |
| IR | **99.6** | 29.6 | 0.4 | **99.5** | 8.6 | 1.1 | 0.1 |
| MBT | **99.0** | 1.0 | 2.2 | 1.8 | **100.0** | 0.1 | 0.0 |
| SA | **98.1** | 0.0 | 0.3 | 0.1 | 0.0 | 16.7 | **100.0** |
| CMA | **99.9** | 8.7 | 0.0 | 0.1 | 0.0 | **97.6** | 5.1 |
| Combination | 94.6 | **99.9** | 100.0 | 99.8 | 100.0 | **99.7** | **100.0** |

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

# A    Broader Impact Statement

Detecting fake images is essential for preserving the quality of internet. Our detectors do not provide certainties, only indications. Our attacks could be used against existing detectors: however, our transfer results (Section 6) show that the benefit of our new detector has more impact as it covers other attacks than the ones specifically considered during the training.

# B    Benchmarks: statistics

| | Min | Max |
|---|---|---|
| Dimension | 1 | 20000 |
| Budget | 10 | 3000000 |
| Num-objectives | 1 | 6 |
| Noise dissymetries | False | True |
| Noise | False | True (many levels) |
| Number of blocks of variables (with independent rotations) | 1 | 16 |
| Number of workers | 1 | 500 |

| Category | Benchmarks |
|---|---|
| Real-world, ML tuning | Keras, Scikit-learn (SVM, Decision Trees, Neural nets) |
| Real-world, not ML tuning | Crops, rockets, energy, fishing, photonics, game |
| Discrete | PBO, Bonnans, others (includes: unordered variables) |

Table 11: Diversity of the benchmarking platform used in our experiments.

Table 11 presents the benchmarks in Nevergrad.

## C  Cases in which log-normal is outperformed by NGOpt

| Cases in which log-normal does not outperform NGOpt | | | |
|---|---|---|---|
| Problem | Rank of LogNormal | Num algorithm | Rank of NGOpt |
| yaonepenboundedbbob | 7 | 86 | 0 |
| yapenboundedbbob | 8 | 87 | 0 |
| yahdbbob | 13 | 72 | 7 |
| naivemltuning | 14 | 21 | 10 |
| reduced-yahdlbbbob | 14 | 88 | 1 |
| yamegapenboundedbbob | 14 | 88 | 8 |
| naive-seq-mltuning | 15 | 23 | 14 |
| seq-mltuning | 17 | 23 | 8 |
| yabbob | 17 | 81 | 11 |
| yaboxbbob | 17 | 53 | 7 |
| nano-naive-veryseq-mltuning | 18 | 27 | 17 |
| instrum-discrete* | 23 | 123 | 4 |
| mltuning | 21 | 23 | 9 |
| yaonepenparabbob | 22 | 56 | 11 |
| double-o-seven | 24 | 39 | 2 |
| yaboundedbbob | 25 | 89 | 4 |
| yapenparabbob | 31 | 56 | 13 |
| yatinybbob | 41 | 89 | 37 |
| zp-ms-bbob | 41 | 58 | 28 |
| ms-bbob | 42 | 65 | 29 |
| powersystems | 42 | 50 | 26 |
| yaparabbob | 46 | 61 | 6 |
| ultrasmall-photonics** | 51 | 85 | 28 |
| mldakmeans | 57 | 70 | 13 |

Table 12: Benchmarks on which log-normal performs weaker than NGOpt. NGOpt is a wizard, automatically choosing an algorithm in a big portfolio of algorithms: it performs vastly better than log-normal for problems with noise (007 and PowerSystems, for which noise management is essential: log-normal can do better for this when combined with Optimism in Front of Uncertainty for dealing with the noise as detailed in Section D.1 and Fig. 2), and for problems derived from BBOB in which high precision by continuous methods is possible.

## D  Black-box Optimization Algorithms

### D.1  List of methods

We briefly present the main BBO algorithms used in the present paper.

- Log-normal (the full name in Nevergrad is LogNormalDiscreteOnePlusOne, sometimes abbreviated as LDOPO), a $(1 + \lambda)$ EA optimizer embedded with the self-adaptive log-normal mutation. This optimizer is adapted from the corresponding discrete version (Kruisselbrink et al., 2011) (see Section 3.1), and details of the tested continuous optimizer are introduced in Section 3.2. It is sometimes abbreviated as LDOPO.

- Lengler (DiscreteLenglerOnePlusone, sometimes abbreviated as Lglr), namely the scheduled mutation rate in (Einarsson et al., 2019).

- Adaptive mutation rates (AdaptiveDiscreteOnePlusOne) come from (Doerr et al., 2019a). This is a $(1 + 1)$ EA with self-adjusting mutation rates. It increases the mutation rate $p$ by $F^s \cdot p$ when the

obtained offspring is at least as good as the parent. Otherwise, $p$ is replaced by $p/F$. $F$ and $s$ are two constant parameters. The algorithm is extended to the continuous optimization following the strategy in Section 3.2.

- Anisotropic discrete algorithms, which uses self-adaptation with a mutation rate per variable.

- NGOpt is a wizard developed by Nevergrad (Rapin & Teytaud, 2018). It automatically chooses an algorithm from a portfolio of algorithms.

- CMA-ES, the Covariance Matrix Adaptation evolution strategy, is a well-known continuous optimization algorithm proposed by Hansen (Hansen & Ostermeier, 2003). We consider CMA and DiagonalCMA (Ros & Hansen, 2008) implemented by Nevergrad (Rapin & Teytaud, 2018) for comparison: the latter is a modified version for high dimensional objective functions.

- Random Search (abbreviated RS) has been commonly applied as a baseline for algorithm comparison. In this paper, we sample new values for each variable uniformly at random.

- MultiSQP, a combination of multiple sequential quadratic programming runs (Artelys, 2015b; Rapin & Teytaud, 2018).

- CMandAS2, a combination of several optimization methods (depending on the dimension and budget, e.g. several CMAs equipped with quadratic MetaModels combined in a bet-and-run(Weise et al., 2018) or the simple $(1 + 1)$ evolution strategy with one-fifth rule(Rechenberg, 1973)) implemented in (Rapin & Teytaud, 2018).

- NGOptRW, the wizard proposed for real-world problems in (Rapin & Teytaud, 2018); it used more DE and more PSO than NGOpt (as opposed to using CMA), and a bet-and-run(Weise et al., 2018).

- Carola2, a chaining (inspired by (Doerr et al., 2018)) between Cobyla (Powell, 1994), CMA (Hansen & Ostermeier, 2003) accompanied by a meta-model and sequential quadratic programming (Artelys, 2015a).

- HyperOpt (Bergstra et al., 2015), based on Parzen estimates.

- NgIoh variants, recent wizards co-developed by the Nevergrad team and the IOH team (Doerr et al., 2018).

- CMA variants: besides DiagonalCMA (Ros & Hansen, 2008), we consider LargeDiagonalCMA (as DiagonalCMA, but with larger initial variance for the population).

- VastDE, basically DE (Storn & Price, 1997) sampling closer to the boundary in bounded cases or with greater variance in unbounded contexts.

- SQOPSO, Special Quasi-Opposite PSO, which adapts quasi-opposite sampling (Rahnamayan et al., 2007) to PSO.

- BAR4, a bet-and-run between

  - quasi-opposite DE folowed by BFGS with finite differences (Rahnamayan et al., 2007) and
  - a CMA equipped with a meta-model followed by a sequential quadratic programming part.

Other methods were run thanks to their availability in the Nevergrad framework; we refer to (Rapin & Teytaud, 2018) for all details.

We use the terminology in the Nevergrad code, i.e., $(1+\lambda)$ optimization methods are derived from the $(1+1)$ code, and contain the suffix OnePlusOne even if $\lambda > 1$.

| Parameter | Value |
|---|---|
| Frequency of update (per iteration) | 1/2 (ZetaSmooth) |
| | 1/3 (UltraSmooth) |
| | 1/9 (SuperSmooth) |
| | 1/55 (Default smooth) |
| Size of smoothing window $s$ | $s = 3$ |

Table 13: Parametrization of the Smooth operator, which operates on a $s \times s$-window.

### D.2 Modifiers of algorithms

### D.2.1 Modifiers dedicated to tensors

When the Smooth operator is applied to a black-box optimization method in Nevergrad, periodically, it tries to replace the current best candidate $x$ by $Smooth(x)$. If $Smooth(x)$ has a better loss value than $x$, then $x$ is replaced by $Smooth(x)$. $Smooth(x)$ is defined as a tensor with the same shape as $x$, with $Smooth(x)_i$ defined as $x_i$ with probability 75%, and, otherwise, the average of the $x_j$ for $j$ the points at distance at most 1 of $i$ in the indices of the tensor $x$. Smooth means that this tentative smoothing is tested once per 55 iterations, SuperSmooth once per 9 iterations; see Table 13.

### D.2.2 Modified dedicated to adversarial attacks

The modifiers G and GSM used for arrays work as follows:

$$
\begin{aligned}
loss_G(x) &= loss(0.03 \times sign(x)) \\
loss_{GSM}(x) &= loss(0.03 \times sign(convolve(x, k)))
\end{aligned}
$$

in the context of an image with values in $[0, 1]$ and an amplitude 0.03 in $l^\infty$. $k$ refers to a Gaussian kernel of width $r/8$ with $r$ the width of the image.

### D.3 Other modifiers of algorithms proposed in Nevergrad

There are other modifiers of algorithms proposed in Nevergrad and visible in prefixes, as follows:

- The prefix SA (self-avoiding, referring to tabu lists) and the suffix Exp (referring to parameters related to simulated annealing) refer to add-ons for the discrete optimization methods.

- Recombining (R for short), which adds a two-point crossover (Holland, 1975) in an algorithm.

- Acceleration by meta-models: by default, MetaModel means CMA plus a quadratic meta-model, but an optimization method (DE, PSO or other) can be specified, and the meta-model can be a random-forest (RF) or a support vector machine (SVM) or a neural network (NN). The algorithm periodically learns a meta-model, and if the learning looks successful it uses the minimum of the MetaModel as a new candidate point.

- Combination with Optimism in front of uncertainty: Nevergrad features bandit tools, which can be combined with other algorithms for making them compatible with noisy optimization. For example, the optimistic counterpart of an algorithm $A$ performs an upper confidence bound method(Lai & Robbins, 1985) for choosing, between points in the search space that have already been used, which one should be resampled, and chooses a new point when the number of function evaluations exceeds a given function of the number of distinct points as specified in (Wang et al., 2009; Rapin & Teytaud, 2018).

## E Significance in Nevergrad plots

In figures created by Nevergrad all dots are independently created. This means, for example,that dots obtained for budget 1000 are not extracted from truncations of runs obtained for budget 2000. This implies

that the robustness of the rankings between methods can be deduced from the stability of curves: if the curve corresponding to algorithm A is always below the curve obtained for Algorithm B (in minimization) this means that Algorithm A performs robustly better than Algorithm B. For computing p-values, the probability that Algorithm A performs better than Algorithm B for the $k$ greatest budget values is at most $1/2^k$ under the null hypothesis that they have the same distribution of average loss values. When computing ranks of algorithms as in Tables 2 and 12, Nevergrad proceeds as follows:

- Compute the average obtained loss $loss_{a,p,b}$ for each algorithm $a$ and each problem $p$ and each budget $b$ (averaged over instances).

- Then the score of $a$ w.r.t algorithm $a'$ is $score_{a,a'}$, frequency at which $loss_{a,p,b} < loss_{a',p,b}$.

- Then, given $n$ the number of algorithms, the score of $a$ is $score_a = \frac{1}{n} \sum_{a'} score_{a,a'}$ and Nevergrad provides the rank of $a$ for this score.

## F   Comparisons of log-normal variants

Figures 2 and 3 compare Lognormal variants on many problems. The optimistic variant is unsurprisingly good for noisy problems and topological problems seem to benefit from anisotropic mutations. Besides that, the standard Lognormal version is never very far from the optimum.

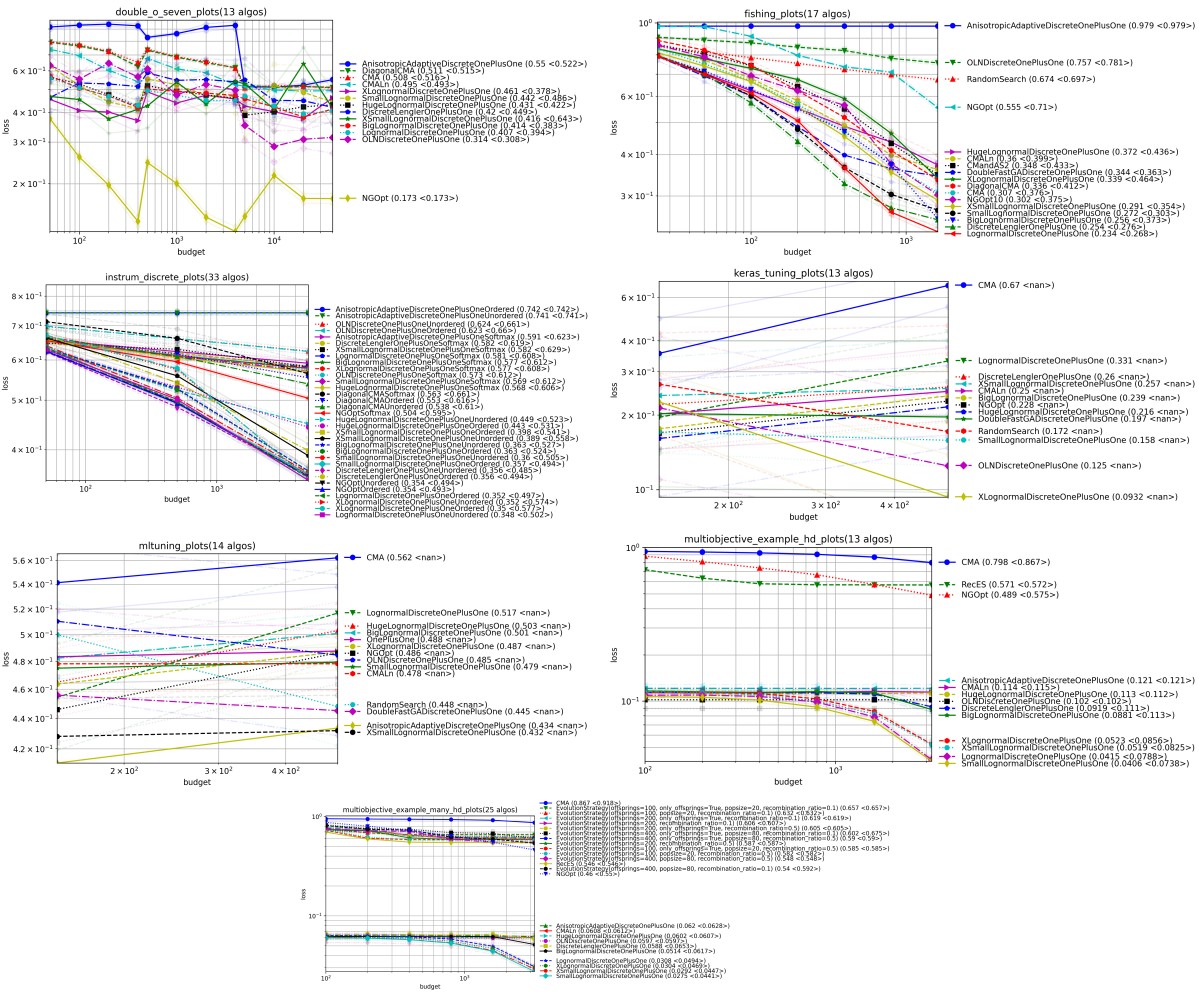

Figure 2: Analysis of variants of Lognormal mutations (1/2). The number between parentheses is the average score for the maximum budget and the number between brackets is the average score for the penultimate budget: as these two figures are obtained in completely independent runs the consistency between both shows the robustness/significance of the ranking (more details in Appendix E). Unsurprisingly, the Optimistic variants (Section D.1) of log-normal algorithms perform well for noisy optimization problems such as 007.

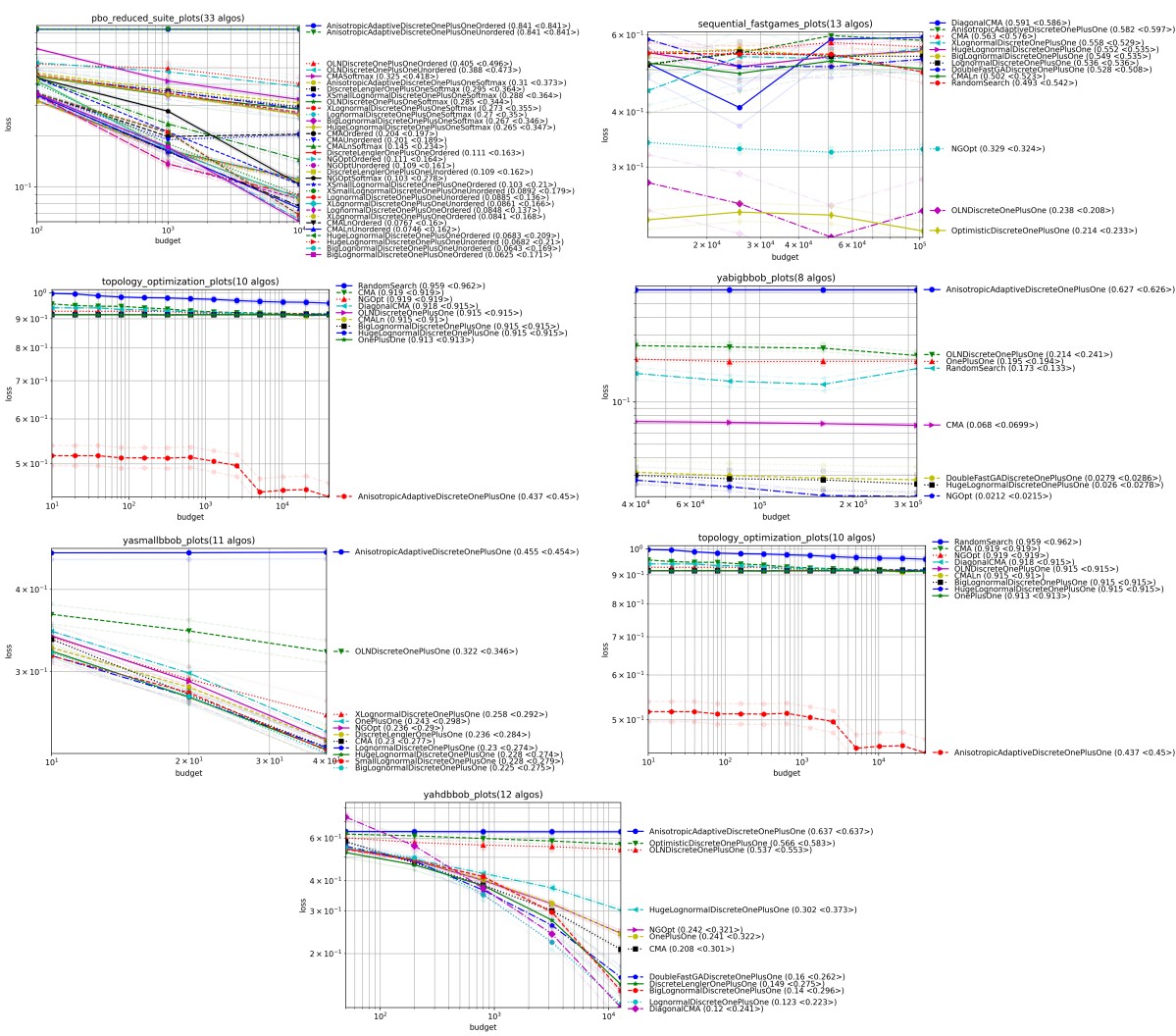

Figure 3: Analysis of variants of Lognormal mutations (2/2). We observe that the default parametrization of LogNormal is essentially ok. Lognormal mutations are a serious competitor in some continuous problems when the prior search distribution used implicitly at initialization and mutation is good and the dimension is high (e.g., YAHDBBOB). For topology optimization, the variable-wise adaptive mutation rate AnisotropicAdaptiveDiscreteOnePlusOne (see Section D.1) is excellent.

## G  Parameters of the learning model

| Hyperparameter | Value |
| --- | --- |
| Type of model | SRnet |
| learning rate | $10^{-4}$ |
| Number of epochs | 20 |
| Hardware | 1 GPU |
| Training time | $\simeq 1$ hour |
| Data Augmentation | JPEG 0.2 prob at 96 quality Resize((256, 256)) RandomHorizontalFlip(0.5) RandomCrop((224, 224)) |

Table 14: Hyper-parameters of our learning model. We work on Dataset2 with each purifier (Dataset2-DP and Dataset2-IR) with parameter 0.1, which is split in 80% train, 10% test, 10% validation.

## H  Full details of the preliminary experiments: LogNormal performs well on Nevergrad benchmarks

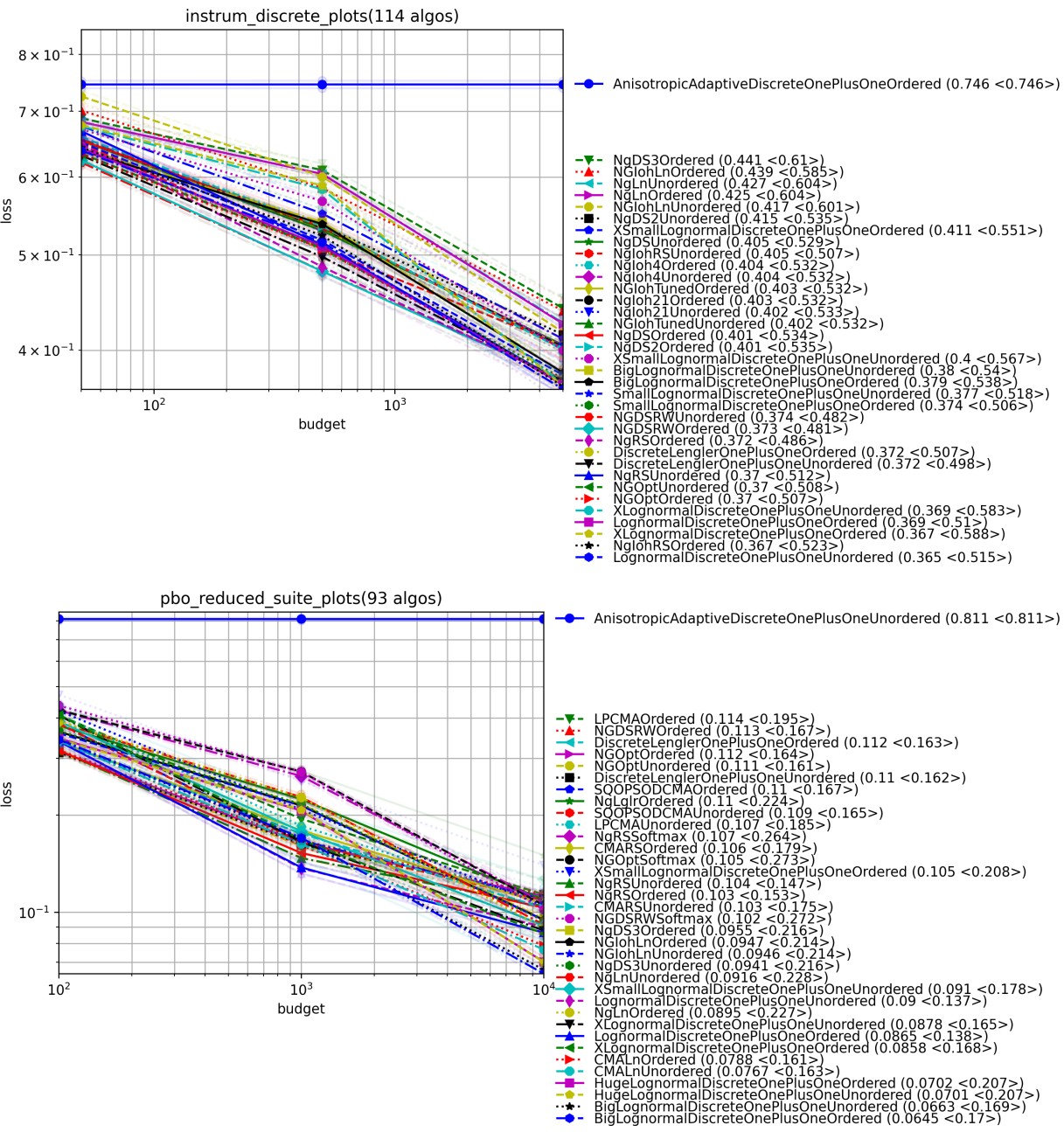

Figure 4: Results on instrum-discrete (top, 35 best methods and the worst method out of 98 methods run on this benchmark), pbo-reduced (bottom, 35 best and the worst method out of 156 methods run on this benchmark). log-normal is simple but good. CMALn is a combination of CMA and log-normal (used as a warmup during the early 10% of the budget): on PBO all strong methods use log-normal at some point.

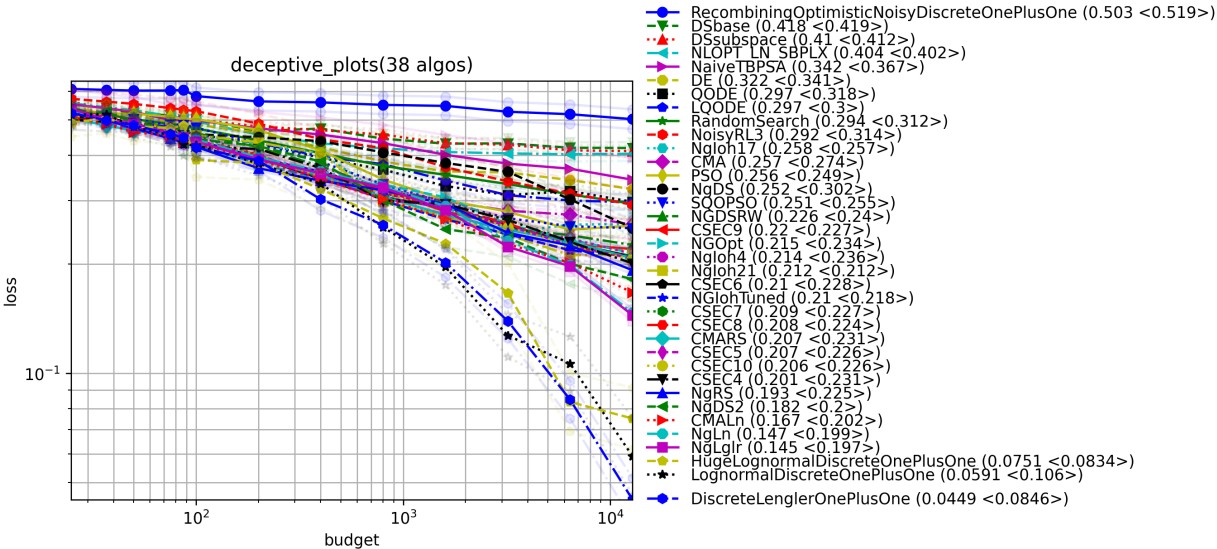

Figure 5: Results on the Deceptive benchmark in Nevergrad. X-axis: budget. Y-axis: average normalized (linearly to $[0, 1]$, for each benchmark) loss. We observe that CMALn (CMA with log-normal warmup) outperforms CMARS (CMA with random search warmup), which outperforms CMA, on this hard benchmark. The best algorithms are based on log-normal: CMALn (resp. NgLn) uses CMA (resp. NGOpt) for local optimization after log-normal. Lengler also performs well, showing that discrete algorithms can be competitive for continuous problems, in the hardest cases, as a warmup or as a standalone method. The CSEC codes are all variants of NgIohTuned: they are good, but still outperformed by codes based on Lognormal. We note an excellent performance of the Lengler method adapted to continuous problems, in particular for the greatest values of the budget, though Table 2 shows that log-normal was better over the different budget values $(25, 37, 50, 75, 87, 100, 200, 400, 800, 1600, 3200, 6400, 12800)$ for the criterion defined in Section E.

