# OpenReview forum: "Lognormal Mutations and their Use in Detecting Surreptitious Fake Images"
_TMLR — Accepted by TMLR_

### Review · Reviewer_6yDZ · 2024-11-25

**Summary Of Contributions:**

This paper proposes a black-box adversarial attack to evade fake image detector. Given the access to the probability scores predicted by the detector model, the proposed evolutionary algorithm with log-normal mutation explores potential adversarial input by randomly mutating the value of some coordinates of the input. Experiments show that this attack can produce images that successfully fool a pre-trained fake image detector and defenses designed for other attacks such as denoising and purification. The authors therefore propose to defense the proposed algorithm by training an additional detector for log-normal attacks.

**Audience:**

Yes

**Claims And Evidence:**

Yes

**Requested Changes:**

Please see concerns above.

**Strengths And Weaknesses:**

**Strengths:**
1. The proposed algorithm is clearly explained and easy to follow. Moreover, it is easy to implement.

2. The problem of identifying and understanding adversarial attacks is critial.


 **Major Concerns:**

1. Effectiveness. Table 4 shows that the baseline method SA achieves a 100\% success rate, which is clearly better than the proposed algorithm, even when the hyperparameters of the proposed algorithm are fine-tuned. It raises my concern how effective the proposed method is. In particular,
   - Why bother using the propose algorithm if an existing method does a better job?
   - While the authors show a list of datasets used in the study, only results on the dataset 1 is reported. Where are the results on other datasets?
   - How to select hyperparameters in an efficient way in practice?
   - Is 10k query budget for each adversarial image or for the whole dataset?
2. Lack of comparisons. There are a number of black-box adversarial attacks developed in the literature besides SA. The authors should add comparison to more baseline methods. Also, the authors should also discuss them as related work.
3. Experiment results. Table 6, 7, 8, 9 has a surprising pattern that the FPR for multiple settings are exactly the same. Can authors explain the reasoning behind?
4. The authors spend a great amount of energy explaining the benefit of choosing log-normal as black-box optimization algorithm. Nevertheless, quoted, "We observe that the log-normal algorithm can outperform NGOpt on 17 out of 41 benchmarks", then why not use NGOpt? Is there any evidence that the deceptive benchmark is strongly connected to the task of creating an adversarial example, or optimizing over the adversarial objective?
5. Contributions. The authors state that they generalize log-normal mutation from discrete case to the continuous case, while in Section 3.2, the authors also acknowledge that "This technique has also been adopted by Nevergrad (Rapin & Teytaud, 2018)". Can authors clarify their contribution on this matter?

**Minor Concerns:**

1. I feel that the introduction part highlights too much on the black-box optimization, while most of them are unrelated to the main content. On the other hand, the introduction and related work on fake image detection is largely ignored. Similarly, Section 2 emphasizes fake detectors but this paper mostly focuses on designing attacks.
2. Figure 1 (and others) is somewhat difficult for the reader to extract useful information since there are too many lines.
3. The proposed defense algorithm is straightforward, namely training a detector on log-normal attacked images.

---

> ### Author Response · Authors · 2024-12-20
> **Answer to comments by reviewer 6yDZ**
>
> We are grateful for the reviewer’s comments, which helps us to clarify our work.
>
> Major Concern 1:
> - The focus of the paper is not on outperforming SA. While SA is very effective, it is also easy to detect, even by visual inspection only. Our core objective is to provide an alternative method that evades detection by SA-specific detectors and any other known detection mechanism.
> - Table 4: Dataset3 is similar to dataset1. Dataset2 and Dataset3-* are datasets created for investigating no-box attacks and black-box attacks, and these are not the ones we want to attack by SA (for making them undetected): SA is about making fake images “look like” (for detectors) real images. Therefore, it is natural to focus on Dataset 1 at this stage. Dataset 1 has a significant diversity in terms of fake images.
> - Selecting hyperparameters: We conclude from preliminaries that the default initial configuration of the Lognormal attack is already sufficiently reasonable, potentially reducing the need for extensive tuning. Moreover, our primary objective is to propose a defense mechanism against Lognormal attacks, and not to focus on one specific attack. The defense, based on supervised learning, performs effectively as demonstrated in Table 9.
> - This is 10k for each image.
>
> Major Concern 2: we agree that there are many other attacks; we show that these attacks are successful (Tables 6 and 7). We show that we can detect an attack (Table 8), which makes them less useful, so that it is important to create new attacks (e.g., Lognormal, Table 4) and then to detect them (Table 9). We are happy to cite more attacks in our revision (changes in red).
>
> Remark 3: the FPR is the same for several rows (i.e. several settings) for a good reason: the False Positive Rate is about negative images, which are the same in those different settings. As you can see in the table, the FPR is not the same when the negative (i.e., real) images are not the same.
>
> Remark 4: we do not want to claim that LogNormal is stronger than NGOpt, and this would be impossible: by design, NGOpt is a wizard, using a carefully chosen algorithm for each context; a new algorithm (unless it is a wizard) is unlikely to outperform NGOpt on average on the entire database of benchmarks in Nevergrad, this would be a big surprise in generic black-box optimization. Our point is more precisely that we are good on problems which have similarities with our target problem, i.e., relatively low budget compared to the dimensionality. It is possible that many other algorithms could be used in lieu of LogNormal: we just point out that this new algorithm (new in the sense that it has never been tested in the Nevergrad benchmarks) does perform well on many benchmarks, in particular, the problems close to adversarial attacks. We need unknown algorithms as attacks, so that we can improve our defense methods: that is why we choose Lognormal, which is somehow new (not available in existing platforms before our work), in particular in the continuous context (to the best of our knowledge, this has never been done, as discussed in point 5 below).
>
> Remark 5: We apologize for the unclear statement: Lognormal was not in Nevergrad before our work, but the tool for extending a discrete optimization method to continuous optimization was already in Nevergrad. Therefore, “Lognormal” in Nevergrad is new, “Lognormal + the extension” is implemented & run for the first time to the best of our knowledge. Only the tool for creating the extension of a discrete method to a continuous setting is not new and was already in Nevergrad.
>
> For clarifying that the key point in our work is to combine (old and new) detectors (including detectors corresponding to new attacks), we add Section 6, and our combined results in Table 10. Also, we move the complicated (hard to read) Fig. 1 in appendix, and rename the corresponding section in the main document as preliminaries: these preliminaries show that, according to the benchmarks in Nevergrad, LogNormal is a reasonable candidate for providing a completely distinct attack.

---

### Review · Reviewer_Ngu7 · 2024-12-04

**Summary Of Contributions:**

The work explores the application of log-normal mutations, a concept originating from evolutionary algorithms, for adversarial attacks and defenses for fake image/data detection. The authors extend the log-normal mutation method to continuous domains and demonstrate its effectiveness in creating adversarial examples that evade detection by existing fake image detectors. Furthermore, the paper introduces new detectors designed to identify these novel attacks, integrating them with fake detection systems to enhance robustness. Extensive benchmarking on the Nevergrad platform and experiments in fake image detection highlight the utility of the proposed method in diverse scenarios.

**Audience:**

No

**Claims And Evidence:**

No

**Requested Changes:**

- Justify why Nevergrad benchmarks are relevant in the context of adversarial attacks/defenses when the majority of the research is conducted in image domain.
- Simplify experimental settings that involve ImageNet and employ a setting similar to [1].
- Report average and median necessary queries.
- Compare the proposed method to SA in low-query regimes.
- Compare the proposed method to some of the other EA methods in image domain.
- Demonstrate experiments with $L_2$ constraints.
- Rewrite the parts of the manuscript where the reader is expected to have large amount of prior knowledge on EAs (in particular, Section 4).

**Strengths And Weaknesses:**

**Strengths**

Extending log-normal mutations from discrete to mixed and continuous domains is an innovative technical contribution.

The paper presents a thorough comparison of log-normal mutations against other optimization algorithms using the Nevergrad benchmarks and performs extensive experiments.

**Weaknesses**

I must say I had a particularly hard time trying to read/understand this manuscript. The work makes strong assumptions about the knowledge of readers about evolutionary algorithms and the experimental suite that's being extensively used. TMLR is not a venue that specifically publishes research on evolutionary algorithms (EAs) and furthermore evolutionary algorithms are not the mainstream line of research for neither adversarial attacks nor defenses. The authors should make a clearer introduction on EAs specifically on this problem and perhaps briefly explain how EAs differ from gradient-based approaches which has been the dominant line of research for adversarial examples. It would also be great if there was a gentle introduction to Nevergrad and perhaps provide some examples for the tested benchmarks. Are these benchmarks even relevant for the adversarial attack/defense community? This is the first time I'm coming across these benchmarks in this context. What are some prior works that use this benchmark in this context and why do they matter?

Figure 1, Figure 2, and Figure 3 are impossible to read. I suggest the authors to provide two versions of these Figures where the first one has every optimization method grayed out except for the proposed one and the second one has everything with colors/icons.

Image datasets used in the paper (provided in Table 3) are too convoluted. I cannot follow the logic of the authors in introducing 11 datasets while only a handful is mentioned/used. I'm comparing this work to the work of [1] which introduces the Square Attack (SA), which is the primary benchmark used in Table 4, and it would be sufficient to simply use Imagenet-1k similar to how it is used in [1]. A simple table such as the Table 2 in [1] which demonstrates success/failure rates on a subset of Imagenet with several models is all that's necessary. I also cannot understand how the authors come to the conclusion provided in the caption of Table 3, which states "The key point for us is that it (the proposed method) performs reasonably well while the transfer from SA detectors fails". SA is denoted to have 100% success rate. How does it fail?

I assume the reason the authors used all other optimization methods for Nevergrad benchmarks and not on ImageNet is because of the automated nature of Nevergrad benchmarking suite. How do (some of) those methods perform on ImageNet compared to the proposed method?

What is the average and the median query necessary for success? The authors limit total queries to 10,000 but that does not mean that all attacks are going to use that many queries. For example, SA also uses 10,000 query limit in their work but average used query (depending on the model) is usually less than 1k.

[1] Andriushchenko et al. Square Attack: a query-efficient black-box adversarial attack via random search

---

> ### Author Response · Authors · 2024-12-20
> **Answer to comments by Reviewer Ngu7, part 1**
>
> We thank the reviewer for the time spent on reviewing our work.
>
> Comment: TMLR is not a venue that specifically publishes research on evolutionary algorithms (EAs) [...]. The authors should make a clearer introduction on EAs specifically on this problem and perhaps briefly explain how EAs differ from gradient-based approaches which has been the dominant line of research for adversarial examples.
>
> Answer:
> We indeed explore the effectiveness of EAs in the context of bypassing fake image detectors. However, we would like to highlight that our study does not solely focus on EAs: we include a lot of black-box optimization methods (see details in appendix).
> Our point is not to find a good adversarial attack in terms of performance for a given L^\infty bound (SA performs very well and we do not need something performing better): our point is to find a decent adversarial attack which is not detected by the detectors of classical attacks. After investigating many methods in the preliminaries, we select one of them for its good performance on problems somewhat related to adversarial attacks, i.e.:  single-objective, low-ratio budget/dimension, deterministic. We rename this section as “Preliminaries” and we move a large part of this discussion to the appendix so that there is no confusion.
> After selecting LogNormal attacks (and other tools could have been used instead of LogNormal, maybe), we switch to the main point of the paper: we show that
> - LogNormal performs reasonably well (though maybe slightly weaker than SA),
> - LogNormal attacks are not detected by detectors trained on SA
> - We can train a detector on generic black-box attacks and then it will detect both SA and new attacks (including LogNormal).
> As requested, we add various references regarding EAs.
>
> Comment: furthermore evolutionary algorithms are not the mainstream line of research for neither adversarial attacks nor defenses.
>
> Answer: The fact that EAs have not been previously explored in the context of adversarial attacks and defense is precisely the important message of our work, for two reasons:
> - LogNormal works surprisingly well on images (though, as mentioned above, we do not claim that it performs better than SA and this is not out point) and has few visual artifacts (making it hard to detect by detectors of classical attacks).
> - Because EAs are very *different* from classical attacks designed specifically for images, we show that classical attack detectors do not work on them (e.g. SA detectors do not work for LogNormal, see Table 8).
> - Therefore, our message is that a general defense should include detectors of generic black-box optimization (and we do this in Tables 9 and 10).
> We are happy to observe (Section 6, Table 10) that a detector trained on only one adversarial attack based on one generic black-box optimization (without LogNormal, for checking transfer), does perform well on LogNormal, thus suggesting some interesting transferability there.

---

> > ### Comment · Reviewer_wxv9 · 2024-12-21
> > **I don't think there's anything EA specific required**
> >
> > In defense of the paper: I read Algorithms 1 and 2 and their are pretty straightforward to understand, no specialized knowledge is required IMHO. I only had concerns with the magnitude of the change, but this was clarified in the revision.

---

> ### Author Response · Authors · 2024-12-20
> **Answer: Part 2**
>
> Comment: Justify why Nevergrad benchmarks are relevant in the context of adversarial attacks/defenses when the majority of the research is conducted in image domain.
>
> Answer: Our aim is to bring new attacks, created in the field of generic black-box optimization, to the domain of adversarial attacks. Whether our choice, namely LogNormal attacks, is the best possible one is not really the question: we use a rather recent method (to the best of our knowledge, lognormal attacks in the continuous domains are applied for the first time in the present paper) and show that it performs reasonably well on a wide range of benchmarks. Then, we switch to the attack of fake detectors.
> For clarifying our purpose, we move the figures about Nevergrad benchmarks in appendix and rename the section about Nevergrad benchmarks as “preliminaries”: that section is just about checking if the LogNormal method is reasonably good and worth testing in the case of adversarial attacks of fake detectors. We also add several parts (in red) clarifying the global goal of the paper:
> - check the state of the art in attacks and detection of attacks, in the context of fake detectors;
> - observe that classical attacks are easily detected (e.g. Table 8);
> - find new attacks, undetected by previous detectors (Table 9 shows that classical detectors do not detect our attack);
> - create new detectors, with some good transfer properties (new Section 6).
>
> Comments requesting more experiments in the original setting of SA: << Simplify experimental settings that involve ImageNet and employ a setting similar to [1]. Compare the proposed method to SA in low-query regimes. Compare the proposed method to some of the other EA methods in image domain. Report average and median necessary queries. Demonstrate experiments with L2  constraints. >>
>
> Answer: We would use the same setting as [1] if our point was to outperform SA. However, our point is to detect attacks, and in particular we want to detect different, alternative attacks. Our results actually show that SA is excellent: the only reason for considering alternative methods is that we want detectors of as many attacks as possible. Our different setup has the advantage of emphasizing the detection, including with transfer from some classes to other classes (see Table 8, showing that classical attacks are detected; Table 9, showing that we can also detect our new attacks; Table 10, showing that our global defense mechanism covers both old and new attacks and also some attacks not used in the training).
>
> The main point of the paper is not to do better than square attacks in terms of success rate. We argue that there are other methods than SA, that they are completely new in the field, and these methods are not detected by detectors of SA (not surprising, after visual inspection in Figure 1: artifacts are completely different). Therefore, we need detectors for other, new, attacks. For clarifying this, we add Section 6, and our combined results in Table 10.
>
> TLDR: We have already included experiments showing that SA is very successful. Outperforming SA is just not the point of our paper: we show the importance of the diversity of attacks, for building a better, robust, defense.
>
> Comment: The work makes strong assumptions about the knowledge of readers about evolutionary algorithms and the experimental suite that's being extensively used.[...] Rewrite the parts of the manuscript where the reader is expected to have large amount of prior knowledge on EAs (in particular, Section 4).
>
> Answer: We believe that understanding most of the paper, and in particular our algorithm, does not require any knowledge of EA nor black-box optimization. Section 4 compares our proposal to other black-box optimization algorithms, and this section indeed needs a background in EA, for which we add references. This section can be skipped by a reader that is only interested in imaging applications. We add a sentence to clarify this. We also rename the section 4 as “preliminaries”, and move many details to the appendix for clarifying what is the main purpose. We also add clarifications (in red in the revision) so that the goal of the paper is not ambiguous.

---

> > ### Comment · Reviewer_Ngu7 · 2025-01-06
> > **On revisions**
> >
> > Thanks for the detailed response to my points. I looked through the changes and the authors' comments, and the majority of my concerns have been resolved.

---

### Review · Reviewer_wxv9 · 2024-12-08

**Summary Of Contributions:**

The paper proposes to apply log-normal mutations to modify fake images (potentially generated by AI image generators). Authors find that these attacks avoid detection by several standard detectors, in particular, these trained to detect a SQUARE attack. Then they show it can be, in turn, mitigated by training detectors on long-normal attack data.

**Audience:**

Yes

**Claims And Evidence:**

Yes

**Requested Changes:**

* Please, introduce the issue of fake images and adversarial attacks on fake image detectors.

* Briefly describe the issue of imperceptible adversarial attacks (listing some common types of attacks on images would be beneficial too).

* Please, clarify whether you use only L_inf attacks. In section 5, you basically provide a definition of an L_inf attack without naming it (so please name it explicitly). However, in algorithm 2, line 6, you seemingly sample a pixel "replacement" from the full domain (i.e., all pixel values).

* What is the exactly the population size lambda? Which population are we talking about here?

* In the end of the paper you write about plans to expand your toolbox with new attacks and defenses. Does it solve all the problems? Please, **shortly** discuss a potential issue of new attacks being invented.

**Strengths And Weaknesses:**

**Strengths**:
* Detection of fake images and robustness of fake image detectors is an important topic
* I have faith in validity of experimental results (i.e., I see no red flags and inconsistencies)
* It is a good demonstration that detectors trained to detect a specific attack can fail in detection of other attack types
* Authors show that their attack is detectable (one can train a detector) even if parameters of the attack used during training are different from those during testing.

**Weaknesses**:
* In adversarial machine learning, we typically only care about imperceptible attacks. SQUARE attack is such an attack (L_inf with eps = 0.03). Is the proposed attack in the same category? I am not sure. In section 5, you basically provide a definition of an L_inf attack without naming it (so please name it explicitly). However, in algorithm 2, line 6, you seemingly sample a pixel "replacement" from the full domain (i.e., all pixel values).
* Given that there can be potentially an infinite number of various attacks, these unseen attacks can potentially evade detection by existing detectors. Once a new attack becomes known, it may (or may not) be possible to train a detector for this attack and expand a list of detectors. But what is an end game for this? I do not expect the paper to solve this problem (possibly unsolvable), but I found no discussion of this issue.
* Although this is not a dealbreaker, the paper is somewhat hard to read. For example, there is no simple introduction/explanation of what is a fake image (and fake detector). I found no definition of no-box attacks. There is no discussion of what is an imperceptible adversarial attack.

---

> ### Author Response · Authors · 2024-12-20
> **Answer to comments by Reviewer wxv9**
>
> We are grateful for your work, which helps us to clarify our paper.
>
> Comment: In adversarial machine learning, we typically only care about imperceptible attacks. SQUARE attack is such an attack (L_inf with eps = 0.03). Is the proposed attack in the same category? I am not sure.
>
> Answer: Thanks for pointing out that this is unclear: yes, we use the exact same norm for all methods, including our LogNormal attack. We are always in $L^\infty$, and the same L^\infty norm is used for all black-box attacks:  we give more details in the answer to the next question.
>
> Comment: In section 5, you basically provide a definition of an L_inf attack without naming it (so please name it explicitly).
>
> Answer: Thanks for pointing out this, we now specify the naming “L_inf”.
>
> Comment: << However, in algorithm 2, line 6, you seemingly sample a pixel "replacement" from the full domain (i.e., all pixel values). >>
>
> Answer: The phrasing was indeed unclear, we clarified it in the revision. This algorithm is written in the general case of optimizing a given objective function in its domain. In our case, the attack is bounded with an L^\infty norm of 0.03, so the domain is [-0.03, 0.03]^(256x256x3) for a 256x256 image with 3 channels: epsilon_{i,j,k} is randomly drawn in [-0.03, 0.03], so that the pixel switches from $v$ to $v+epsilon_{i,j,k}$. Similarly, for other benchmarks (in particular the general purpose benchmarks), the value is randomly drawn in the domain of the corresponding variable, as specified in each benchmark (we specify this in the revision).
> For more clarity, we add section 3.3 with an overview of the LogNormal attack and of its footprint on images.
>
> Comment: Given that there can be potentially an infinite number of various attacks, these unseen attacks can potentially evade detection by existing detectors. Once a new attack becomes known, it may (or may not) be possible to train a detector for this attack and expand a list of detectors. But what is an end game for this? I do not expect the paper to solve this problem (possibly unsolvable), but I found no discussion of this issue.
>
> Answer: We agree that this is important. To clarify this, we add Section 6 with a table of results, namely Table 10. This shows that a detector (row “Combination”) combining several detectors can detect the lognormal attack and the SA attack. The key point, as explained in our (new) text, is that this detector was trained on another attack created from generic black-box optimization, namely the CMA black-box optimization method: this shows a reasonably good transfer between different attacks. We agree that there is more work to do on this, and mention it as further work: but this is a first step and we do have transfer between different attacks.
>
> Comment: Although this is not a dealbreaker, the paper is somewhat hard to read. For example, there is no simple introduction/explanation of what is a fake image (and fake detector). I found no definition of no-box attacks. There is no discussion of what is an imperceptible adversarial attack.
>
> Answer: We added definitions of no-box and imperceptible adversarial attacks to the main text.
>
> Comment: Please, introduce the issue of fake images and adversarial attacks on fake image detectors.
> Added to the text of the main paper, in the introduction (all modifications are in red).
> Briefly describe the issue of imperceptible adversarial attacks (listing some common types of attacks on images would be beneficial too).
>
> Answer: Thanks: done in section 5.1.
>
> Comment: Please, clarify whether you use only L_inf attacks. In section 5, you basically provide a definition of an L_inf attack without naming it (so please name it explicitly). However, in algorithm 2, line 6, you seemingly sample a pixel "replacement" from the full domain (i.e., all pixel values).
>
> Answer: Done, thanks. Modifications appear in red in our revision.
>
> Comment: What is the exactly the population size lambda? Which population are we talking about here?
>
> Answer: This is about parallelism: we clarify by stating this in the tex.
>
> Comment: In the end of the paper you write about plans to expand your toolbox with new attacks and defenses. Does it solve all the problems? Please, shortly discuss a potential issue of new attacks being invented.
>
> Answer: Yes this matters. We add a short comment in the broader impact statement.

---

> > ### Comment · Reviewer_wxv9 · 2024-12-21
> > **concerns resolved**
> >
> > Thank you! I looked through the changes and my concerns are resolved. The paper focuses only on standard L_inf (not easily perceptible) attacks and now it is explained more clearly in the paper.

---

### Comment · Action_Editor_PS4y · 2025-02-21
**Issues with author list**

Dear authors,
There's a problem with the formatting of your author list. Can you look at a published article at TMLR to see the error? Somehow the author tex macros are being misused and the affiliations are appearing in the wrong place.

---

> ### Comment · Action_Editor_PS4y · 2025-02-23
> **Notice.**
>
> Notice that this paper cannot proceed to publication until the mis-formatting is fixed.

---

> > ### Author Response · Authors · 2025-02-24
> > **Fixing the author information.**
> >
> > Thank you.
> > We fixed this issue. The new version is uploaded.

---

### Decision · Action_Editor_PS4y · 2025-01-15

**Recommendation:** Accept as is

**Comment:**

There were two key concerns: technical soundness and clarity. Both have been adddressed to the reviewers' satisfication.

1. Technical Soundness. The experimental results were validated by multiple reviewers, with updates such as clarifying Linf norm constraints and detailing the log-normal mutation’s mechanics enhancing rigor.

2 Clarity and Accessibility. Definitions of key terms, a restructured introduction, and simplified experimental settings addressed Reviewer Ngu7’s concerns about accessibility. Moving ancillary benchmarking results to the appendix has simpliefied the main paper.

Relevance and Broader Implications: The authors demonstrated their approach onreal-world adversarial detection scenarios while acknowledging limitations. The inclusion of new detectors trained on generic black-box attacks provides transferable insights for detecting novel attack methods.

In summary, the paper makes a clear contribution to adversarial robustness in fake detection, and its interdisciplinary approach broadens the scope of tools available for addressing this important problem.

**Audience:**

The paper’s findings will likely be of broad interest, and in particular to the subset of TMLR’s audience focused on adversarial robustness. Reviewer wxv9 emphasized the relevance of adversarial robustness in fake image detection, and this paper contributes a novel perspective by leveraging methods from evolutionary algorithms.

Although Reviewer Ngu7 questioned the prominence of evolutionary algorithms in this context, the authors clarified their broader purpose: to explore methods outside mainstream approaches and highlight their potential in adversarial contexts. The revisions included simplified benchmarks and datasets to enhance accessibility and contextual relevance. These changes mean the paper speaks more effectively to both adversarial ML researchers and those exploring novel optimization techniques.

**Claims And Evidence:**

The claims made in the submission are reasonably well supported by experimental results and the revisions addressing reviewer concerns.

Reviewer wxv9 expressed confidence in the validity of the experiments and noted no methodological problems. The authors strengthened the presentation by clarifying the use of Linf bounded attacks, explicitly naming them in Section 5 and updating Algorithm 2 to highlight the constraints.

Reviewer Ngu7 raised concerns about the clarity and accessibility of the manuscript, in particular its reliance on prior knowledge of evolutionary algorithms. To address this, the authors restructured the introduction and Section 3, providing context on evolutionary algorithms and distinguishing them from gradient-based methods. These changes explain better the role of the log-normal mutation and its relevance to adversarial attacks.

Additionally, Table 10 and Section 6 demonstrate transferability of detectors trained on generic black-box optimization attacks to detect new attacks, such as those using log-normal mutations. This added evidence strengthens the paper's main claims and provides practical insights for improving fake detection robustness. That said, I agree with the comment that "I am not sure that defending against this additional attack makes us much more secure because potentially other attacks can be invented." This is the typical problem but now we have a new attack to pay attention to.